# Diffusion-Reward Adversarial Imitation Learning

**Chun-Mao Lai**[1][*]  **Hsiang-Chun Wang**[1][*]  **Ping-Chun Hsieh**[2]  **Yu-Chiang Frank Wang**[1,3]
**Min-Hung Chen**[3]  **Shao-Hua Sun**[1]
[1]National Taiwan University  [2]National Yang Ming Chiao Tung University  [3]NVIDIA

## Abstract

Imitation learning aims to learn a policy from observing expert demonstrations without access to reward signals from environments. Generative adversarial imitation learning (GAIL) formulates imitation learning as adversarial learning, employing a generator policy learning to imitate expert behaviors and discriminator learning to distinguish the expert demonstrations from agent trajectories. Despite its encouraging results, GAIL training is often brittle and unstable. Inspired by the recent dominance of diffusion models in generative modeling, we propose Diffusion-Reward Adversarial Imitation Learning (DRAIL), which integrates a diffusion model into GAIL, aiming to yield more robust and smoother rewards for policy learning. Specifically, we propose a diffusion discriminative classifier to construct an enhanced discriminator, and design diffusion rewards based on the classifier's output for policy learning. Extensive experiments are conducted in navigation, manipulation, and locomotion, verifying DRAIL's effectiveness compared to prior imitation learning methods. Moreover, additional experimental results demonstrate the generalizability and data efficiency of DRAIL. Visualized learned reward functions of GAIL and DRAIL suggest that DRAIL can produce more robust and smoother rewards. Project page: https://nturobotlearninglab.github.io/DRAIL/

## 1  Introduction

Imitation learning, *i.e.*, learning from demonstration [24, 41, 49], aims to acquire an agent policy by observing and mimicking the behavior demonstrated in expert demonstrations. Various imitation learning methods [53, 60] have enabled deploying reliable and robust learned policies in a variety of tasks involving sequential decision-making, especially in the scenarios where devising a reward function is intricate or uncertain [7, 32, 34], or when learning in a trial-and-error manner is expensive or unsafe [14, 17].

Among various methods in imitation learning, generative adversarial imitation learning (GAIL) [21] has been widely adopted due to its effectiveness and data efficiency. GAIL learns a generator policy to imitate expert behaviors through reinforcement learning and a discriminator to differentiate between the expert and the generator's state-action pair distributions, resembling the idea of generative adversarial networks (GANs) [16]. Despite its established theoretical guarantee, GAIL training is notoriously brittle and unstable. To alleviate this issue, significant efforts have been put into improving GAIL's sample efficiency, scalability, robustness, and generalizability by modifying loss functions [12], developing improved policy learning algorithms [30], and exploring various similarity measures of distributions [2, 8, 12].

Inspired by the recent dominance of diffusion models in generative modeling [22], this work explores incorporating diffusion models into GAIL to provide more robust and smoother reward functions

---

[*]Equal contribution.  Correspondence to: Shao-Hua Sun <shaohuas@ntu.edu.tw>

38th Conference on Neural Information Processing Systems (NeurIPS 2024).

for policy learning as well as stabilize adversarial training. Specifically, we propose a diffusion discriminative classifier, which learns to classify a state-action pair into expert demonstrations or agent trajectories with merely two reverse diffusion steps. Then, we leverage the proposed diffusion discriminative classifier to devise diffusion rewards, which reward agent behaviors that closely align with expert demonstrations. Putting them together, we present Diffusion-Reward Adversarial Imitation Learning (DRAIL), a novel adversarial imitation learning framework that can efficiently and effectively produce reliable policies replicating the behaviors of experts.

We extensively compare our proposed framework DRAIL with behavioral cloning [44], Diffusion Policy [6, 42], and AIL methods, *e.g.*, GAIL [21], WAIL [2], and DiffAIL[59], in diverse continuous control domains, including navigation, robot arm manipulation, locomotion, and games. This collection of tasks includes environments with high-dimensional continuous state and action spaces, as well as covers both vectorized and image-based states. The experimental results show that our proposed framework consistently outperforms the prior methods or achieves competitive performance. Moreover, DRAIL exhibits superior performance in generalizing to states or goals unseen from the expert's demonstrations. When varying the amounts of available expert data, DRAIL demonstrates the best data efficiency. At last, the visualized learned reward functions show that DRAIL captures more robust and smoother rewards compared to GAIL.

## 2 Related work

Imitation learning enables agents to learn from expert demonstrations to acquire complex behaviors without explicit reward functions. Its application spans various domains, including robotics [49, 60], autonomous driving [36], and game AI [18].

**Behavioral Cloning (BC).** BC [44, 55] imitates an expert policy through supervised learning without interaction with environments and is widely used for its simplicity and effectiveness across various domains. Despite its benefits, BC struggles to generalize to states not covered in expert demonstrations because of compounding error [10, 48]. Recent methods have explored learning diffusion models as policies [6, 42], allowing for modeling multimodal expert behaviors, or using diffusion models to provide learning signals to enhance the generalizability of BC [5]. In contrast, this work aims to leverage a diffusion model to provide learning signals for policy learning in online imitation learning.

**Inverse Reinforcement Learning (IRL).** IRL methods [37] aim at inferring a reward function that could best explain the demonstrated behavior and subsequently learn a policy using the inferred reward function. Nevertheless, inferring reward functions is an ill-posed problem since different reward functions could induce the same demonstrated behavior. Hence, IRL methods often impose constraints on reward functions or policies to ensure optimality and uniqueness [1, 37, 54, 61]. Yet, these constraints could potentially restrict the generalizability of learned policies.

**Adversarial Imitation Learning (AIL).** AIL methods aim to directly match the state-action distributions of an agent and an expert through adversarial training. Generative adversarial imitation learning (GAIL) [21] and its extensions [23, 25, 31, 56, 59, 62] train a generator policy to imitate expert behaviors and a discriminator to differentiate between the expert and the generator's state-action pair distributions, which resembles the idea of generative adversarial networks (GANs) [16]. Thanks to its simplicity and effectiveness, GAIL has been widely applied to various domains [3, 28, 45]. Over the past years, researchers have proposed numerous improvements to enhance GAIL's sample efficiency, scalability, and robustness [40], including modifications to discriminator's loss function [12], extensions to off-policy RL algorithms [30], addressing reward bias [31], and exploration of various similarity measures [2, 8, 11, 12]. Another line of work avoids adversarial training, such as IQ-Learn [15], which learns a Q-function that implicitly represents the reward function and policy. In this work, we propose to use the diffusion model as a discriminator in GAIL.

**Diffusion Model-Based Approaches in Reinforcement Learning.** Diffuser [57] and Nuti et al. [39] apply diffusion models to reinforcement learning (RL) and reward learning, their settings differ significantly from ours. Diffuser [57] is a model-based RL method that requires trajectory-level reward information, which differs from our setting, *i.e.*, imitation learning, where obtaining rewards is not possible. Nuti et al. [39] focus on learning a reward function, unlike imitation learning, whose goal is to obtain a policy. Hence, Nuti et al. [39] neither present policy learning results in the main paper nor compare their method to imitation learning methods. Moreover, they focus on learning

from a fixed suboptimal dataset, AIL approaches and our method are designed to learn from agent data that continually change as the agents learn.

## 3 Preliminaries

We propose a novel adversarial imitation learning framework that integrates a diffusion model into generative adversarial imitation learning. Hence, this section presents background on the two topics.

### 3.1 Generative Adversarial Imitation Learning (GAIL)

GAIL [21] establishes a connection between generative adversarial networks (GANs) [16] and imitation learning. GAIL employs a generator, $G_\theta$, that acts as a policy $\pi_\theta$, mapping a state to an action. The generator aims to produce a state-action distribution ($\rho_{\pi_\theta}$) which closely resembles the expert state-action distribution $\rho_{\pi_E}$; discriminator $D_\omega$ functions as a binary classifier, attempting to differentiate the state-action distribution of the generator ($\rho_{\pi_\theta}$) from the expert's ($\rho_{\pi_E}$). The optimization equation of GAIL can be formulated using the Jensen-Shannon divergence, which is equivalent to the minimax equation of GAN. The optimization of GAIL can be derived as follows:

$$\min_\theta \max_\omega \mathbb{E}_{\mathbf{x}\sim\rho_{\pi_\theta}}[\log D_\omega(\mathbf{x})] + \mathbb{E}_{\mathbf{x}\sim\rho_{\pi_E}}[\log(1 - D_\omega(\mathbf{x}))], \tag{1}$$

where $\rho_{\pi_\theta}$ and $\rho_{\pi_E}$ are the state-action distribution from an agent $\pi_\theta$ and expert policy $\pi_E$ respectively. The loss function for the discriminator is stated as $-(\mathbb{E}_{\mathbf{x}\sim\rho_{\pi_\theta}}[\log D_\omega(\mathbf{x})] + \mathbb{E}_{\mathbf{x}\sim\rho_{\pi_E}}[\log(1 - D_\omega(\mathbf{x}))])$. For a given state, the generator tries to take expert-like action; the discriminator takes state-action pairs as input and computes the probability of the input originating from an expert. Then the generator uses a reward function $-\mathbb{E}_{\mathbf{x}\sim\rho_{\pi_\theta}}[\log D_\omega(\mathbf{x})]$ or $-\mathbb{E}_{\mathbf{x}\sim\rho_{\pi_\theta}}[\log D_\omega(\mathbf{x})] + \lambda H(\pi_\theta)$ to optimize its network parameters, where the entropy term $H$ is a policy regularizer controlled by $\lambda \geq 0$.

### 3.2 Diffusion models

Diffusion models have demonstrated state-of-the-art performance on various tasks [9, 29, 38, 51]. This work builds upon denoising diffusion probabilistic models (DDPMs) [22] that employ forward and reverse diffusion processes, as illustrated in Figure 1. The forward diffusion process injects noise into data points following a variance schedule until achieving an isotropic Gaussian distribution. The reverse diffusion process trains a diffusion model $\phi$ to predict the injected noise by optimizing the objective:

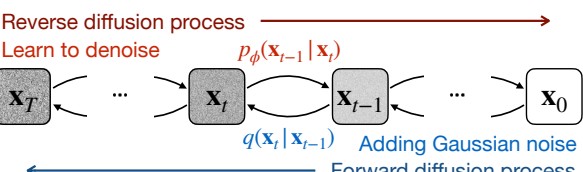

Figure 1: **Denoising diffusion probabilistic model.** Latent variables $x_1, ..., x_N$ are produced from the data point $x_0$ via a forward diffusion process, *i.e.*, gradually adding noises to the latent variables. A diffusion model $\phi$ learns to reverse the diffusion process by denoising the noisy data to reconstruct the original data point $x_0$.

$$\mathcal{L}_{DM} = \mathbb{E}_{t\sim T, \boldsymbol{\epsilon}\sim\mathcal{N}}\left[\left\|\boldsymbol{\epsilon}_\phi(\sqrt{\bar{\alpha}_t}\mathbf{x}_0 + \sqrt{1 - \bar{\alpha}_t}\boldsymbol{\epsilon}, t) - \boldsymbol{\epsilon}\right\|^2\right], \tag{2}$$

where $T$ represents the set of discrete time steps in the diffusion process, $\boldsymbol{\epsilon}$ is the noise applied by the forward process, $\boldsymbol{\epsilon}_\phi$ is the noise predicted by the diffusion model, and $\bar{\alpha}_t$ is the scheduled noise level applied on the data samples.

Beyond generative tasks, diffusion models have also been successfully applied in other areas, including image classification and imitation learning. Diffusion Classifier [35] demonstrates that conditional diffusion models can estimate class-conditional densities for zero-shot classification. In imitation learning, diffusion models have been used to improve Behavioral Cloning (DBC) [5] by using diffusion model to model expert state-action pairs. Similarly, DiffAIL [58] extends GAIL [21] by employing diffusion models to represent the expert's behavior and incorporating the diffusion loss into the discriminator's learning process. However, DiffAIL's use of an unconditional diffusion model limits its ability to distinguish between expert and agent state-action pairs. We provide a detailed explanation of its limitation in Section 4.3 and Section A.

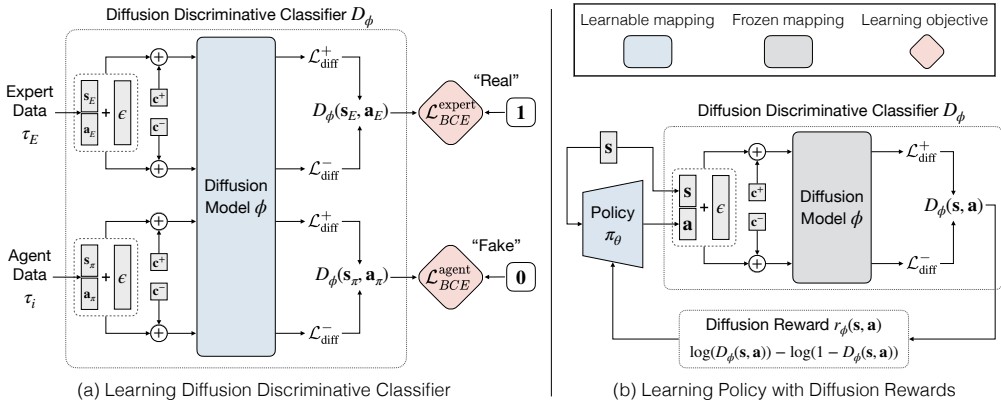

Figure 2: **Diffusion-Reward Adversarial Imitation Learning.** Our proposed framework DRAIL incorporates a diffusion model into GAIL. **(a)** Our proposed diffusion discriminative classifier $D_\phi$ learns to distinguish expert data $(\mathbf{s}_E, \mathbf{a}_E) \sim \tau_E$ from agent data $(\mathbf{s}_\pi, \mathbf{a}_\pi) \sim \tau_i$ using a diffusion model. $D_\phi$ is trained to predict a value closer to 1 when the input state-action pairs are sampled from expert demonstration and predict a value closer to 0 otherwise. **(b)** The policy $\pi_\theta$ learns to maximize the diffusion reward $r_\phi$ computed based on the output of $D_\phi$ that takes the state-action pairs from the policy as input. The closer the policy resembles expert behaviors, the higher the rewards it can obtain.

## 4 Approach

We propose a novel adversarial imitation learning framework incorporating diffusion models into the generative adversarial imitation learning (GAIL) framework, illustrated in Figure 2. Specifically, we employ a diffusion model to construct an enhanced discriminator to provide more robust and smoother rewards for policy learning. We initiate our discussion by describing a naive integration of the diffusion model, which directly predicts rewards from Gaussian noises conditioned on state-action pairs, and the inherent issues of this method in Section 4.1. Subsequently, in Section 4.2, we introduce our proposed method that employs a conditional diffusion model to construct a diffusion discriminative classifier, which can provide diffusion rewards for policy learning. Finally, the overall algorithm of our method is outlined in Section 4.3.

### 4.1 Reward prediction with a conditional diffusion model

Conditional diffusion models are widely adopted in various domains, *e.g.*, generating an image $x$ from a label $y$. Intuitively, one can incorporate a conditional diffusion model as a GAIL discriminator by training it to produce a real or fake label conditioned on expert or agent state-action pairs. Specifically, given a denoising time step $t$ and a state-action pair $(\mathbf{s}, \mathbf{a}) \in (\mathcal{S} \times \mathcal{A})$, where $\mathcal{S}, \mathcal{A}$ stand for state and action spaces, respectively, as a condition, the diffusion model $p_\phi(r_{t-1}|r_t, \mathbf{s}, \mathbf{a})$ learns to denoise a reward label $r_0 \in \{0, 1\}$, *i.e.*, 1 for expert (real) state-action pairs and 0 for agent (fake) state-action pairs through a reverse diffusion process.

To train a policy, we can use the diffusion model to produce a reward $r$ given a state-action pair $(\mathbf{s}, \mathbf{a})$ from the policy through a generation process by iteratively denoising a sampled Gaussian noise, *i.e.*, noisy reward, conditioned on the state-action pair. Then, the policy learns to optimize the rewards predicted by the diffusion model. Nevertheless, the reward generation process is extremely time-consuming since predicting a reward for each state-action pair from the policy requires running $T$ (often a large number) denoising steps, and policy learning often takes tens of millions of samples, resulting in a billion-level overall training scale. Consequently, it is impractical to integrate a diffusion model into the GAIL framework by using it to predict "realness" rewards for policy learning from state-action pairs.

### 4.2 Diffusion discriminative classifier

Our goal is to yield a diffusion model reward given an agent state-action pair without going through the entire diffusion generation process. Inspired by previous work [5, 59], we extract the learning signal from a portion of the diffusion denoising steps, rather than using the entire process. Building

on these insights, we adapt the training procedure of DDPM to develop a mechanism that provides a binary classification signal using just one denoising step.

Our key insight is to leverage the derivations developed by Kingma et al. [26], Song et al. [52], which suggest that the diffusion loss, *i.e.*, the difference between the predicted noise and the injected noise, indicates how well the data fits the target distribution since the diffusion loss is the upper bound of the negative log-likelihood of data in the target distribution. In this work, we propose calculating "realness" rewards based on the diffusion loss computed by denoising the state-action pairs from the policy, which indicates how well the state-action pairs fit the expert behavior distributions. We formulate the diffusion loss $\mathcal{L}_{\mathrm{diff}}$ as follows:

$$\mathcal{L}_{\mathrm{diff}}(\mathbf{s}, \mathbf{a}, \mathbf{c}) = \mathbb{E}_{t \sim T}\left[\|\boldsymbol{\epsilon}_\phi(\mathbf{s}, \mathbf{a}, \boldsymbol{\epsilon}, t|\mathbf{c}) - \boldsymbol{\epsilon}\|^2\right], \tag{3}$$

where $\mathbf{c} \in \{\mathbf{c}^+, \mathbf{c}^-\}$, and the real label $\mathbf{c}^+$ corresponds to the condition for fitting expert data while the fake label $\mathbf{c}^-$ corresponds to agent data. We implement $\mathbf{c}^+$ as $\mathbf{1}$ and $\mathbf{c}^-$ as $\mathbf{0}$.

To approximate the expectation in Eq. 3, we use random sampling, allowing us to achieve the result with just a single denoising step. Subsequently, given a state-action pair $(\mathbf{s}, \mathbf{a})$, $\mathcal{L}_{\mathrm{diff}}(\mathbf{s}, \mathbf{a}, \mathbf{c}^+)$ measures how well $(\mathbf{s}, \mathbf{a})$ fits the expert distribution and $\mathcal{L}_{\mathrm{diff}}(\mathbf{s}, \mathbf{a}, \mathbf{c}^-)$ measures how well $(\mathbf{s}, \mathbf{a})$ fits the agent distribution[1]. That said, given state-action pairs sampled from expert demonstration, $\mathcal{L}_{\mathrm{diff}}^+$ should be close to 0, and $\mathcal{L}_{\mathrm{diff}}^-$ should be a large value; on the contrary, given agent state-action pairs, $\mathcal{L}_{\mathrm{diff}}^+$ should be a large value and $\mathcal{L}_{\mathrm{diff}}^-$ should close to 0.

While $\mathcal{L}_{\mathrm{diff}}^+$ and $\mathcal{L}_{\mathrm{diff}}^-$ can indicate the "realness" or the "fakeness" of a state-action pair to some extent, optimizing a policy using rewards with this wide value range $[0, \infty)$ can be difficult [20]. To address this issue, we propose transforming this diffusion model into a binary classifier that provides "realness" in a bounded range of $[0, 1]$. Specifically, given the diffusion model's output $\mathcal{L}_{\mathrm{diff}}^{+;-}$, we construct a diffusion discriminative classifier $D_\phi : \mathcal{S} \times \mathcal{A} \to \mathbb{R}$:

$$D_\phi(\mathbf{s}, \mathbf{a}) = \frac{e^{-\mathcal{L}_{\mathrm{diff}}(\mathbf{s},\mathbf{a},\mathbf{c}^+)}}{e^{-\mathcal{L}_{\mathrm{diff}}(\mathbf{s},\mathbf{a},\mathbf{c}^+)} + e^{-\mathcal{L}_{\mathrm{diff}}(\mathbf{s},\mathbf{a},\mathbf{c}^-)}} = \sigma(\mathcal{L}_{\mathrm{diff}}(\mathbf{s}, \mathbf{a}, \mathbf{c}^-) - \mathcal{L}_{\mathrm{diff}}(\mathbf{s}, \mathbf{a}, \mathbf{c}^+)), \tag{4}$$

where $\sigma(x) = 1/(1 + e^{-x})$ denotes the sigmoid function. The classifier integrates $\mathcal{L}_{\mathrm{diff}}^+$ and $\mathcal{L}_{\mathrm{diff}}^-$ to compute the "realness" of a state-action pair within a bounded range of $[0, 1]$, as illustrated in Figure 2. Since the design of our diffusion discriminative classifier aligns with the GAIL discriminator [21], learning a policy with the classifier enjoys the same theoretical guarantee, *i.e.*, optimizing this objective can bring a policy's occupancy measure closer to the expert's. Consequently, we can optimize our proposed diffusion discriminative classifier $D_\phi$ with the loss function:

$$\mathcal{L}_D = \mathbb{E}_{(\mathbf{s},\mathbf{a})\in\tau_E} \underbrace{[-\log(D_\phi(\mathbf{s}, \mathbf{a}))]}_{\mathcal{L}_{BCE}^{\mathrm{expert}}} + \mathbb{E}_{(\mathbf{s},\mathbf{a})\in\tau_i} \underbrace{[-\log(1 - D_\phi(\mathbf{s}, \mathbf{a}))]}_{\mathcal{L}_{BCE}^{\mathrm{agent}}} \tag{5}$$

where $\mathcal{L}_D$ sums the expert binary cross-entropy loss $\mathcal{L}_{BCE}^{\mathrm{expert}}$ and the agent binary cross-entropy loss $\mathcal{L}_{BCE}^{\mathrm{agent}}$, and $\tau_E$ and $\tau_i$ represent a sampled expert trajectory and a collected agent trajectory by the policy $\pi$ at training step $i$. We then update the diffusion discriminative classifier parameters $\phi$ based on the gradient of $\mathcal{L}_D$ to improve its ability to distinguish expert data from agent data.

Intuitively, the discriminator $D_\phi$ is trained to predict a value closer to $1$ when the input state-action pairs are sampled from expert demonstration (*i.e.*, trained to minimize $\mathcal{L}_{\mathrm{diff}}^+$ and maximize $\mathcal{L}_{\mathrm{diff}}^-$), and $0$ if the input state-action pairs are obtained from the agent online interaction (*i.e.*, trained to minimize $\mathcal{L}_{\mathrm{diff}}^-$ and maximize $\mathcal{L}_{\mathrm{diff}}^+$).

Note that our idea of transforming the diffusion model into a classifier is closely related to Li et al. [35], which shows that minimizing the diffusion loss is equivalent to maximizing the evidence lower bound (ELBO) of the log-likelihood [4], allowing for turning a conditional text-to-image diffusion model into an image classifier by using the ELBO as an approximate class-conditional log-likelihood $\log p(x|c)$. By contrast, we employ a diffusion model for imitation learning. Moreover, we take a step further – instead of optimizing the diffusion loss $\mathcal{L}_{\mathrm{diff}}$, we directly optimize the binary cross entropy losses calculated based on the denoising results to train the diffusion model as a binary classifier.

---

[1] For simplicity, we will use the notations $\mathcal{L}_{\mathrm{diff}}^+$ and $\mathcal{L}_{\mathrm{diff}}^-$ to represent $\mathcal{L}_{\mathrm{diff}}(\mathbf{s}, \mathbf{a}, \mathbf{c}^+)$ and $\mathcal{L}_{\mathrm{diff}}(\mathbf{s}, \mathbf{a}, \mathbf{c}^-)$, respectively, in the rest of the paper. Additionally, $\mathcal{L}_{\mathrm{diff}}^{+;-}$ denotes $\mathcal{L}_{\mathrm{diff}}^+$ and $\mathcal{L}_{\mathrm{diff}}^-$ given a state-action pair.

### 4.3 Diffusion-Reward Adversarial Imitation Learning

Our proposed method adheres to the fundamental AIL framework, where a discriminator and a policy are updated alternately. In the discriminator step, we update the diffusion discriminative classifier with the gradient of $\mathcal{L}_D$ following Eq. 5. In the policy step, we adopt the adversarial inverse reinforcement learning objective proposed by Fu et al. [12] as our diffusion reward signal to train the policy:

$$r_\phi(\mathbf{s}, \mathbf{a}) = \log(D_\phi(\mathbf{s}, \mathbf{a})) - \log(1 - D_\phi(\mathbf{s}, \mathbf{a})). \quad (6)$$

The policy parameters $\theta$ can be updated using any RL algorithm to maximize the diffusion rewards provided by the diffusion discriminative classifier, bringing the policy closer to the expert policy. In our implementation, we utilize PPO as our policy update algorithm. The algorithm is presented in Algorithm 1, and the overall framework is illustrated in Figure 2.

---

**Algorithm 1** Diffusion-Reward Adversarial Imitation Learning (DRAIL)

---

1: **Input:** Expert trajectories $\tau_E$, initial policy parameters $\theta_0$, initial diffusion discriminator parameters $\phi_0$, and discriminator learning rate $\eta_\phi$
2: **for** $i = 0, 1, 2, \ldots$ **do**
3:    Sample agent transitions $\tau_i \sim \pi_{\theta_i}$
4:    Compute the output of diffusion discriminative classifier $D_\phi$ (Eq. 4) and the loss function $\mathcal{L}_D$ (Eq. 5)
5:    Update the diffusion model $\phi_{i+1} \leftarrow \phi_i - \eta_\phi \nabla \mathcal{L}_D$
6:    Compute the diffusion reward $r_\phi(\mathbf{s}, \mathbf{a})$ with Eq. 6
7:    Update the policy $\theta_{i+1} \leftarrow \theta_i$ with any RL algorithm w.r.t. reward $r_\phi$
8: **end for**

---

Among the related works, DiffAIL [59] is the closest to ours, as it also uses a diffusion model for adversarial imitation learning. DiffAIL employs an unconditional diffusion model to denoise state-action pairs from both experts and agents. However, this approach only implicitly reflects the likelihood of state-action pairs belonging to the expert class through diffusion loss, making it challenging to explicitly distinguish between expert and agent behaviors.

In contrast, our method, DRAIL, uses a conditional diffusion model that directly conditions real ($c^+$) and fake ($c^-$) labels. This allows our model to explicitly calculate and compare the probabilities of state-action pairs belonging to either the expert or agent class. This clearer and more robust signal for binary classification aligns more closely with the objectives of the GAIL framework, leading to more stable and effective learning. For further details and the mathematical formulation, please refer to Section A.

## 5 Experiments

We extensively evaluate our proposed framework DRAIL in diverse continuous control domains, including navigation, robot arm manipulation, and locomotion. We also examine the generalizability and data efficiency of DRAIL in Section 5.4 and Section 5.5. The reward function learned by DRAIL is presented in Section 5.6.

### 5.1 Experimental setup

This section describes the environments, tasks, and expert demonstrations used for evaluation.

**MAZE.** We evaluate our approach in the point mass MAZE navigation environment, introduced in Fu et al. [13] (maze2d-medium-v2), as depicted in Figure 3a. In this task, a point-mass agent is trained to navigate from a randomly determined start location to the goal. The agent accomplishes the task by iteratively predicting its acceleration in the vertical and horizontal directions. We use the expert dataset provided by Lee et al. [33], which includes 100 demonstrations, comprising 18,525 transitions.

**FETCHPUSH.** We evaluate our approach in a 7-DoF Fetch task, FETCHPUSH, depicted in Figure 3b, where the Fetch is required to push a black block to a designated location marked by a red sphere. We use the demonstrations from Lee et al. [33], consisting of 20,311 transitions (664 trajectories).

**HANDROTATE.** We further evaluate our approach in a challenging environment HANDROTATE with a *high-dimensional continuous action space* introduced by Plappert et al. [43]. Here, a 24-DoF Shadow Dexterous Hand is tasked with learning to in-hand rotate a block to a target orientation, as depicted in Figure 3c. This environment features a high-dimensional state space (68D) and action space (20D). We use the demonstrations collected by Lee et al. [33], which contain 515 trajectories (10k transitions).

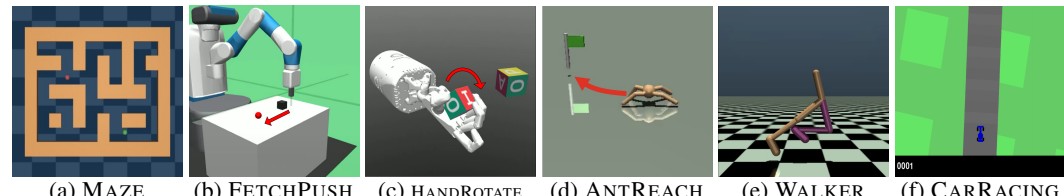

(a) MAZE    (b) FETCHPUSH    (c) HANDROTATE    (d) ANTREACH    (e) WALKER    (f) CARRACING

Figure 3: **Environments & tasks. (a) MAZE**: A point-mass agent (green) within a 2D maze is trained to move from its initial position to reach the goal (red). **(b) FETCHPUSH**: The manipulation task is implemented with a 7-DoF Fetch robotics arm. FETCHPUSH requires picking up or pushing an object to a target location (red). **(c) HANDROTATE**: For this dexterous manipulation task, a Shadow Dexterous Hand is employed to in-hand rotate a block to achieve a target orientation. **(d) ANTREACH**: This task trains a quadruped ant to reach a goal randomly positioned along the perimeter of a half-circle with a radius of 5 m. **(e) WALKER**: This locomotion task requires training a bipedal walker policy to achieve the highest possible walking speed while maintaining balance. **(f) CARRACING** This image-based racing game task requires driving a car to navigate a track as quickly as possible.

**ANTREACH.** The goal of ANTREACH, a location and navigation task, is for a quadruped ant to reach a goal randomly positioned along the perimeter of a half-circle with a radius of 5 meters, as depicted in Figure 3d. The 132D *high-dimensional continuous state space* encodes joint angles, velocities, contact forces, and the goal position relative to the agent. We use the demonstrations provided by Lee et al. [33], which contain 1,000 demonstrations (25k transitions).

**WALKER.** The objective of WALKER is to let a bipedal agent move at the highest speed possible while preserving its balance, as illustrated in Figure 3e. We trained a PPO expert policy with environment rewards and collected 5 successful trajectories, each containing 1000 transitions, as an expert dataset.

**CARRACING.** We evaluate our method in a racing game, CARRACING, illustrated in Figure 3f, requiring driving a car to navigate a track. This task features a $96 \times 96$ RGB *image-based state space* and a 3-dimensional action space (steering, braking, and accelerating). We trained a PPO expert policy on CARRACING environment and collected 671 transitions as expert demonstrations.

Further details of the tasks can be found in Section B.

## 5.2 Baselines

We compare our method DRAIL with the following baselines of our approach.

- **Behavioral Cloning (BC)** trains a policy to mimic the actions of an expert by supervised learning a mapping from observed states to corresponding expert actions [44, 55].
- **Diffusion Policy** represents a policy as a conditional diffusion model [6, 42, 46], which predicts an action conditioning on a state and a randomly sampled noise. We include this method to compare learning a diffusion model as a *policy* (diffusion policy) or *reward function* (ours).
- **Generative Adversarial Imitation Learning (GAIL)** [21] learns a policy from expert demonstrations by training a discriminator to distinguish between trajectories generated by the learned generator policy and those from expert demonstrations.
- **Generative Adversarial Imitation Learning with Gradient Penalty (GAIL-GP)** is an extension of GAIL that introduces a gradient penalty to achieve smoother rewards and stabilize the discriminator.
- **Wasserstein Adversarial Imitation Learning (WAIL)** [2] extends GAIL by employing Wasserstein distance, aiming to capture smoother reward functions.
- **Diffusion Adversarial Imitation Learning (DiffAIL)** [58] integrates a diffusion model into AIL by using the diffusion model loss to provide reward $e^{-\mathcal{L}_{\text{diff}}}$.

## 5.3 Experimental results

We present the success rates (MAZE, FETCHPUSH, HANDROTATE, ANTREACH) and return (WALKER, CARRACING) of all the methods with regards to environment steps in Figure 4. Each task

was trained using five different random seeds. Note that BC and Diffusion Policy are offline imitation learning algorithms, meaning they cannot interact with the environment, so their performances are represented as horizontal lines. Detailed information on model architectures, training, and evaluation can be found in Section F and Section G.

Overall, our method DRAIL consistently outperforms prior methods or achieves competitive performance compared to the best-performing methods across all the environments, verifying the effectiveness of integrating our proposed diffusion discriminative classifier into the AIL framework.

**DRAIL vs. DiffAIL.** Both DRAIL and DiffAIL integrate the diffusion model into the AIL framework. In 5 out of 6 tasks, our DRAIL outperforms DiffAIL, demonstrating that our proposed discriminator provides a more effective learning signal by closely resembling binary classification within the GAIL framework.

**DRAIL vs. BC.** AIL methods generally surpass BC in most tasks due to their ability to learn from interactions with the environment and thus handle unseen states better. However, BC outperforms all other baselines in the locomotion task (WALKER). We hypothesize that WALKER is a monotonic task requiring less generalizability to unseen states, allowing BC to excel with sufficient expert data. Additionally, our experiments with varying amounts of expert data, detailed in Section 5.5, suggest that DRAIL surpasses BC when less expert data is available.

We empirically found that our proposed DRAIL is robust to hyperparameters, especially compared to GAIL and WAIL, as shown in the hyperparameter sensitivity experiment inSection D.

## 5.4 Generalizability

To examine the generalizability to states or goals that are unseen from the expert demonstrations of different methods, we extend the FETCHPUSH tasks following the setting proposed by Lee et al. [33]. Specifically, we evaluate policies learned by different methods by varying the noise injected into initiate states (*e.g.*, position and velocity of the robot arm) and goals (*e.g.*, target block positions in FETCHPUSH). We experiment with different noise levels, including $1\times$, $1.25\times$, $1.5\times$, $1.75\times$, and $2.0\times$, compared to the expert environment. That said, $1.5\times$ means the policy is evaluated in an environment with noises $1.5\times$ larger than those injected into expert data collection. Performing well in a high noise level setup requires the policy to generalize to unseen states.

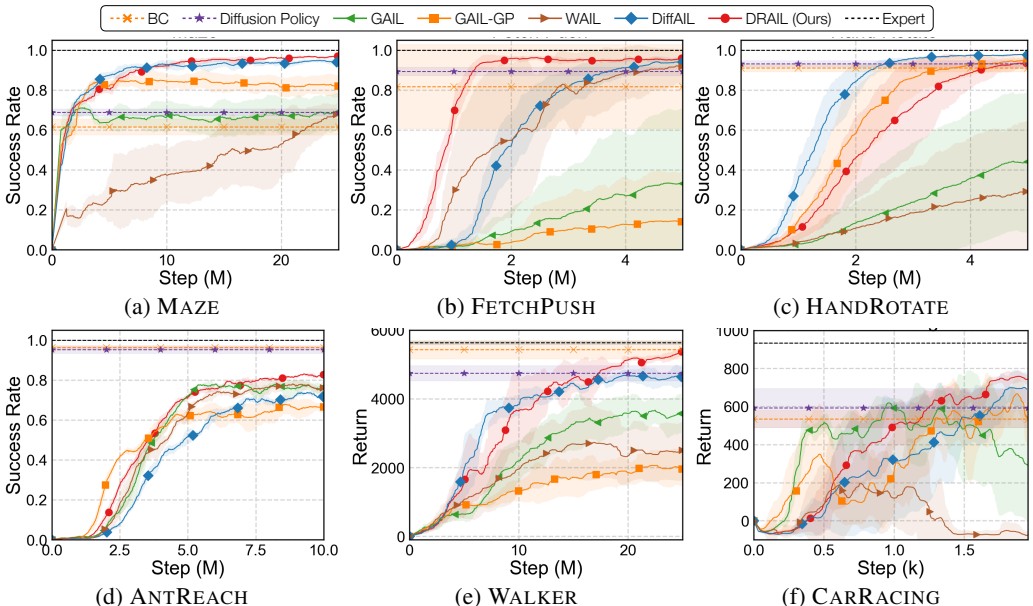

Figure 4: **Learning efficiency.** We report success rates (MAZE, FETCHPUSH, HANDROTATE, ANTREACH) and return (WALKER, CARRACING), evaluated over five random seeds. Our method DRAIL learns more stably, faster, and achieves higher or competitive performance compared to the best-performing baseline in all the tasks.

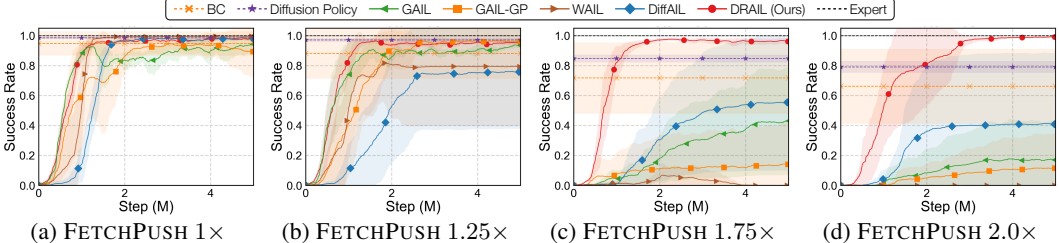

(a) FETCHPUSH $1\times$    (b) FETCHPUSH $1.25\times$    (c) FETCHPUSH $1.75\times$    (d) FETCHPUSH $2.0\times$

Figure 5: **Generalization experiments in FETCHPUSH.** We present the performance of our proposed DRAIL and baselines in the FETCHPUSH task, under varying levels of noise in initial states and goal locations. The evaluation spans three random seeds, and the training curve illustrates the success rate dynamics.

The results of FETCHPUSH under $1\times$, $1.25\times$, $1.75\times$, and $2.0\times$ noise level are presented in Figure 5. Across all noise levels, approaches utilizing the diffusion model generally exhibit better performance. Notably, our proposed DRAIL demonstrates the highest robustness towards noisy environments. Even at the highest noise level of $2.00\times$, DRAIL maintains a success rate of over $95\%$, surpassing the best-performing baseline, Diffusion Policy, which achieves only around a $79.20\%$ success rate. In contrast, DiffAIL experiences failures in 2 out of the 5 seeds, resulting in a high standard deviation (mean: $40.90$, standard deviation: $47.59$), despite our extensive efforts on experimenting with various configurations and a wide range of hyperparameter values.

The extended generalization experiments results in MAZE, FETCHPUSH, HANDROTATE , and the new task FETCHPICK are presented in Section C. Overall, our method outperforms or performs competitively against the best-performing baseline, demonstrating its superior generalization ability.

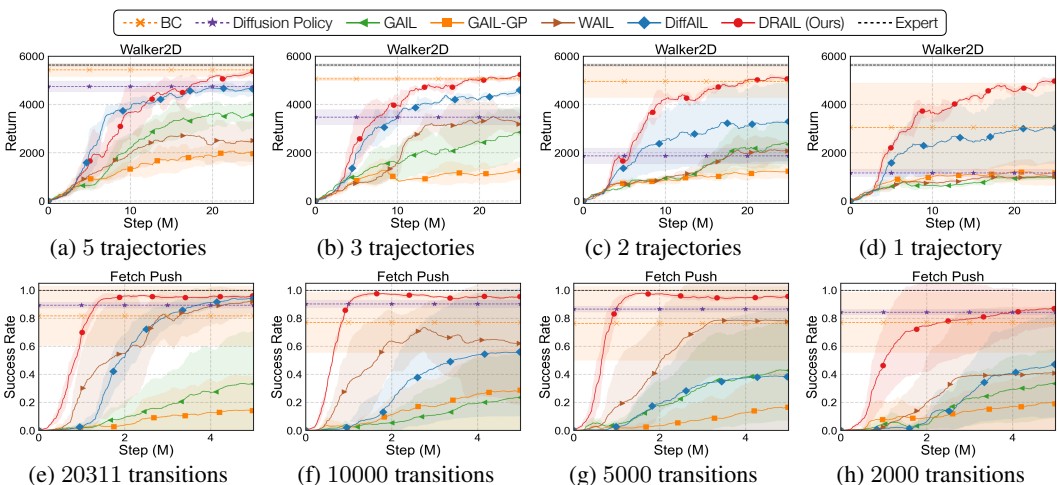

(a) 5 trajectories    (b) 3 trajectories    (c) 2 trajectories    (d) 1 trajectory

(e) 20311 transitions    (f) 10000 transitions    (g) 5000 transitions    (h) 2000 transitions

Figure 6: **Data efficiency.** We experiment learning with varying amounts of expert data in WALKER and FETCHPUSH. The results show that our proposed method DRAIL is more data efficient, *i.e.*, can learn with less expert data, compared to other methods.

## 5.5 Data efficiency

To investigate the data efficiency of DRAIL, we vary the amount of expert data used for learning in WALKER and FETCHPUSH. Specifically, for WALKER, we use 5, 3, 2, and 1 expert trajectories, each containing 1000 transitions; for FETCHPUSH, we use 20311, 10000, 5000, and 2000 state-action pairs. The results reported in Figure 6 demonstrate that our DRAIL learns faster compared to the other baselines, indicating superior data efficiency in terms of environment interaction. In WALKER, our DRAIL maintains a return value of over 4500 even when trained with a single trajectory. In contrast, BC's performance is unstable and fluctuating, while the other baselines experience a dramatic drop. In FETCHPUSH, our DRAIL maintains a success rate of over $80\%$ even when the data size is reduced by $90\%$, whereas the other AIL baselines' performance drops below $50\%$.

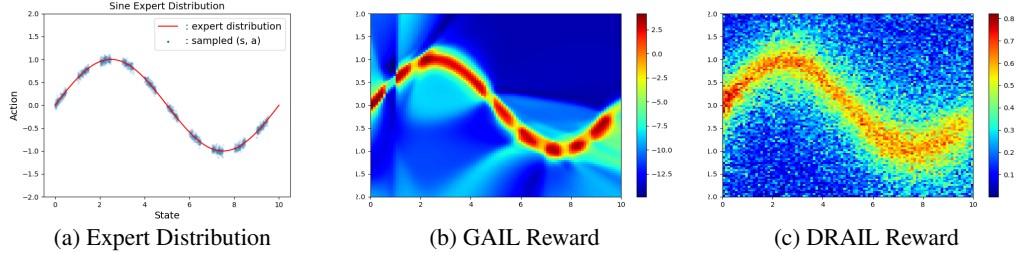

|  (a) Expert Distribution | (b) GAIL Reward | (c) DRAIL Reward |

Figure 7: **Reward function visualization.** We present visualizations of the learned reward values by the discriminative classifier of GAIL and the diffusion discriminative classifier of our DRAIL. The target expert demonstration for imitation is depicted in **(a)**, which is a discontinuous sine function. The reward distributions of GAIL and our DRAIL are illustrated in **(b)** and **(c)**, respectively.

## 5.6 Reward function visualization

To visualize and analyze the learned reward functions, we design a SINE environment with one-dimensional state and action spaces, where the expert state-action pairs form a discontinuous sine wave $\mathbf{a} = \sin(20\mathbf{s}\pi) + \mathcal{N}(0, 0.05^2)$, as shown in Figure 7a. We train GAIL and our DRAIL to learn from this expert state-action distribution and visualize the discriminator output values $D_\phi$ to examine the learned reward function, as presented in Figure 7.

Figure 7b reveals that the GAIL discriminator exhibits excessive overfitting to the expert demonstration, resulting in its failure to provide appropriate reward values when encountering unseen states. In contrast, Figure 7c shows that our proposed DRAIL generalizes better to the broader state-action distribution, yielding a more robust reward value, thereby enhancing the generalizability of learned policies. Furthermore, the predicted reward value of DRAIL gradually decreases as the state-action pairs deviate farther from the expert demonstration. This reward smoothness can guide the policy even when it deviates from the expert policy. In contrast, the reward distribution from GAIL is relatively narrow outside the expert demonstration, making it challenging to properly guide the policy if the predicted action does not align with the expert.

## 6 Conclusion

This work proposes a novel adversarial imitation learning framework that integrates a diffusion model into generative adversarial imitation learning (GAIL). Specifically, we propose a diffusion discriminative classifier that employs a diffusion model to construct an enhanced discriminator, yielding more robust and smoother rewards. Then, we design diffusion rewards based on the classifier's output for policy learning. Extensive experiments in navigation, manipulation, locomotion, and game justify our proposed framework's effectiveness, generalizability, and data efficiency. Future work could apply DRAIL to image-based robotic tasks in real-world or simulated environments and explore its potential in various domains outside robotics, such as autonomous driving, to assess its generalizability and adaptability. Additionally, exploring other divergences and distance metrics, such as the Wasserstein distance or f-divergences, could potentially further improve training stability.

## Acknowledgement

This work was supported by NVIDIA Taiwan Research & Development Center (TRDC) under the funding code 112HT911007. Shao-Hua Sun was supported by the Yushan Fellow Program by the Ministry of Education, Taiwan. Ping-Chun Hsieh is supported in part by the National Science and Technology Council (NSTC), Taiwan under Contract No. NSTC 113-2628-E-A49-026. We thank Tim Pearce and Ching-An Cheng for the fruitful discussions.

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

# Appendix

# Table of Contents

## A  Relation to DiffAIL

Among all the related works, DiffAIL [58], which also employs a diffusion model for adversarial imitation learning, is the closest to our work. This section describes the differences of our work and DiffAIL.

**Unconditional diffusion models (DiffAIL) vs. conditional diffusion models (DRAIL).** Wang et al. [58] proposed to learn an *unconditional* diffusion model $D_\phi(\mathbf{s}, \mathbf{a})$ to denoise state-action pairs from an expert and an agent. The diffusion model should denoise well when the state-action pairs are sampled from the expert demonstrations, while denoise poorly given the state-action pairs sampled from the agent policy. On the other hand, our proposed framework employs a *conditional* diffusion model $D_\phi(\mathbf{s}, \mathbf{a}, \mathbf{c})$ that conditions on the real label $\mathbf{c}^+$ or the fake label $\mathbf{c}^-$. In the following, we discuss how using a conditional diffusion model leads to better learning signals for policy learning.

DiffAIL set up the discriminator output as $D_\phi(\mathbf{s}, \mathbf{a}) = e^{-\mathcal{L}_{\text{diff}}(\mathbf{s}, \mathbf{a})} \in [0, 1]$, indicating how likely $(\mathbf{s}, \mathbf{a})$ is sampled from the expert distribution. That said, a $e^{-\mathcal{L}_{\text{diff}}(\mathbf{s}, \mathbf{a})}$ close to 1 implies that the sample is likely from expert and a small value of $e^{-\mathcal{L}_{\text{diff}}(\mathbf{s}, \mathbf{a})}$ means it is less likely that this sample

comes from the expert. However, $e^{-\mathcal{L}_{\text{diff}}(\mathbf{s},\mathbf{a})}$ does not explicitly represent the probability of the "negative class" (agent), which makes it difficult to provide stable rewards for policy learning.

We start by revisiting the GAIL formulation. The binary classifier in GAIL, *i.e.*, the discriminator, is trained to minimize the binary cross-entropy loss, which leads to maximizing the likelihood of predicting the correct class probabilities. Subsequently, the discriminator outputs the predicted probability of the "positive class" (expert) relatively to the "negative class" (agent). In DiffAIL, the predicted probability of the "positive class" (expert) is $e^{-\mathcal{L}_{\text{diff}}}$ and the predicted probability of the "negative class" (agent) can be given by $1 - e^{-\mathcal{L}_{\text{diff}}}$. Following this definition, the discriminator thinks the sample if from an expert when $e^{-\mathcal{L}_{\text{diff}}} > 1 - e^{-\mathcal{L}_{\text{diff}}}$. We also know that $\mathcal{L}_{\text{diff}} > 0$ since it is the diffusion model denoising loss. This gives us $0 \leq \mathcal{L}_{\text{diff}} \leq \ln 2$ for the expert class, and $\ln 2 < \mathcal{L}_{\text{diff}}$ for the agent class. This can be problematic since the fixed boundary $\ln 2$ my not reflect the real boundary between the real (expert) and the fake (agent) distributions.

On the other hand, in our method (DRAIL), the predicted probability of the positive (expert) class of a given state-action pair $(\mathbf{s}, \mathbf{a})$ is defined as:

$$D_\phi(\mathbf{s}, \mathbf{a}) = \frac{e^{-\mathcal{L}_{\text{diff}}(\mathbf{s},\mathbf{a},\mathbf{c}^+)}}{e^{-\mathcal{L}_{\text{diff}}(\mathbf{s},\mathbf{a},\mathbf{c}^+)} + e^{-\mathcal{L}_{\text{diff}}(\mathbf{s},\mathbf{a},\mathbf{c}^-)}} = \sigma(\mathcal{L}_{\text{diff}}(\mathbf{s}, \mathbf{a}, \mathbf{c}^-) - \mathcal{L}_{\text{diff}}(\mathbf{s}, \mathbf{a}, \mathbf{c}^+)), \qquad (7)$$

and the predicted probability of the negative (agent) class of the same state-action pair $(\mathbf{s}, \mathbf{a})$ is explicitly defined as:

$$1 - D_\phi(\mathbf{s}, \mathbf{a}) = \frac{e^{-\mathcal{L}_{\text{diff}}(\mathbf{s},\mathbf{a},\mathbf{c}^-)}}{e^{-\mathcal{L}_{\text{diff}}(\mathbf{s},\mathbf{a},\mathbf{c}^+)} + e^{-\mathcal{L}_{\text{diff}}(\mathbf{s},\mathbf{a},\mathbf{c}^-)}} = \sigma(\mathcal{L}_{\text{diff}}(\mathbf{s}, \mathbf{a}, \mathbf{c}^+) - \mathcal{L}_{\text{diff}}(\mathbf{s}, \mathbf{a}, \mathbf{c}^-)). \qquad (8)$$

This binary classification formulation aligns with GAIL. Hence, the discriminator would think a given $(\mathbf{s}, \mathbf{a})$ comes from expert data only when

$$\frac{e^{-\mathcal{L}_{\text{diff}}(\mathbf{s},\mathbf{a},\mathbf{c}^+)}}{e^{-\mathcal{L}_{\text{diff}}(\mathbf{s},\mathbf{a},\mathbf{c}^+)} + e^{-\mathcal{L}_{\text{diff}}(\mathbf{s},\mathbf{a},\mathbf{c}^-)}} > \frac{e^{-\mathcal{L}_{\text{diff}}(\mathbf{s},\mathbf{a},\mathbf{c}^-)}}{e^{-\mathcal{L}_{\text{diff}}(\mathbf{s},\mathbf{a},\mathbf{c}^+)} + e^{-\mathcal{L}_{\text{diff}}(\mathbf{s},\mathbf{a},\mathbf{c}^-)}}, \qquad (9)$$

which means

$$e^{-\mathcal{L}_{\text{diff}}(\mathbf{s},\mathbf{a},\mathbf{c}^+)} > e^{-\mathcal{L}_{\text{diff}}(\mathbf{s},\mathbf{a},\mathbf{c}^-)}. \qquad (10)$$

The resulting relative boundary can provide better learning signals for policy learning, especially when the behaviors of the agent policy become similar to those of expert, which can be observed in the tasks where the agent policies can closely follow the experts, such as FETCHPICK, FETCHPUSH, and ANTREACH. Also, we hypothesize this leads to the superior performance of our method compared to DiffAIL in most of the generalization experiments.

**Experimental setup.** Wang et al. [58] evaluated DiffAIL and prior methods in locomotion tasks; in contrast, our work extensively compares our proposed framework DRAIL with various existing methods in various domains, including navigation (MAZE and ANTREACH), locomotion (WALKER and ANTREACH), robot arm manipulation (FETCHPUSH and FETCHPICK), robot arm dexterous (HANDROTATE), and games (CARRACING). Additionally, we present experimental results on generalization to unseen states and goals on the goal-oriented tasks, and on varying amounts of expert data.

# B  Environment & task details

## B.1  MAZE

**Description.** In a 2D maze environment, a point-maze agent learns to navigate from a starting location to a goal location. The agent achieves this by iteratively predicting its x and y velocity. The initial and final positions of the agent are randomly selected. The state space includes position,

velocity, and goal position. The maximum episode length for this task is set at 400, and the episode terminates if the goal is reached earlier.

**Expert dataset.** The expert dataset comprises 100 demonstrations, which includes $18, 525$ transitions provided by Lee et al. [33].

## B.2 FETCHPUSH & FETCHPICK

**Description.** In the FETCHPUSH task, the agent is required to push an object to a specified target location. On the other hand, in the FETCHPICK task, the objective is to pick up an object from a table and move it to a target location.

According to the environment setups stated in Lee et al. [33], the 16-dimensional state representation includes the angles of the robot joints, and the initial three dimensions of the action vector represent the intended relative position for the next time step. The first three dimensions of the action vector denote the intended relative position in the subsequent time step. In the case of FETCHPICK, an extra action dimension is incorporated to specify the distance between the two fingers of the gripper. The maximum episode length for this task is set at 60 for FETCHPUSH and 50 for FETCHPICK, and the episode terminates if the agent reaches the goal earlier.

**Expert dataset.** The expert dataset for FETCHPUSH comprises 664 trajectories, amounting to $20, 311$ transitions, and the expert dataset for FETCHPICK comprises 303 trajectories, amounting to $10, 000$ transitions provided by Lee et al. [33].

## B.3 HANDROTATE

**Description.** In the task HANDROTATE proposed by Plappert et al. [43], a 24-DoF Shadow Dexterous Hand is designed to rotate a block in-hand to a specified target orientation. The 68D state representation includes the agent's joint angles, hand velocities, object poses, and target rotation. The 20D action vector represents the joint torque control of the 20 joints. Notably, HANDROTATE is challenging due to its high-dimensional state and action spaces. We follow the experimental setup outlined in Plappert et al. [43] and Lee et al. [33], where rotation is constrained to the z-axis, and allowable initial and target z rotations are within $\left[-\frac{\pi}{12}, \frac{\pi}{12}\right]$ and $\left[\frac{\pi}{3}, \frac{2\pi}{3}\right]$, respectively. The maximum episode length for this task is set at $50$, and the episode terminates if the hand reaches the goal earlier.

**Expert dataset.** We use the demonstrations collected by Lee et al. [33], which contain 515 trajectories (10k transitions).

## B.4 ANTREACH

**Description.** The ANTREACH task features a four-leg ant robot reaching a randomly assigned target position located within a range of half-circle with a radius of 5 meters. The task's state is represented by a 132-dimension vector, including joint angles, velocities, and the relative position of the ant towards the goal. Expert data collection for this task is devoid of any added noise, while random noise is introduced during the training and inference phases. Consequently, the policy is required to learn to generalize to states not present in the expert demonstrations. The maximum episode length for this task is set at 50, and the episode terminates if the ant reaches the goal earlier.

**Expert dataset.** The expert dataset comprises 10000 state-action pairs provided by Lee et al. [33].

## B.5 WALKER

**Description.** WALKER task involves guiding an agent to move towards the x-coordinate as fast as possible while maintaining balance. An episode terminates either when the agent experiences predefined unhealthy conditions in the environment or when the maximum episode length (1000) is reached. The agent's performance is evaluated over 100 episodes with three different random seeds. The return of an episode is the cumulative result of all time steps within that episode. The 17D state includes joint angles, angular velocities of joints, and velocities of the x and z-coordinates of the top. The 6D action defines the torques that need to be applied to each joint of the walker avatar.

**Expert dataset.** We trained a PPO expert policy with environment rewards and collected 5 successful trajectories, each containing 1000 transitions, as an expert dataset.

## B.6 CARRACING

**Description.** In the CARRACING task, the agent must navigate a track by controlling a rear-wheel drive car. The state space of CARRACING is represented by a top-down $96 \times 96$ RGB image capturing the track, the car, and various status indicators such as true speed, four ABS sensors, steering wheel position, and gyroscope readings. The agent controls the car using three continuous action values: steering, acceleration, and braking. Episodes have a maximum length of 1000 steps, and termination occurs if the car completes the track before reaching the maximum episode length.

In our experiment settings, we preprocess the state image by converting it to grayscale and resizing it to $64 \times 64$ pixels. We then concatenate two consecutive frames to form a single state data, providing temporal context for the agent's decision-making process.

**Expert dataset.** We trained a PPO expert policy on CARRACING environment and collected 671 transitions as expert demonstrations.

### B.7 Expert performance

For MAZE, FETCHPUSH, FETCHPICK, HANDROTATE, and ANTREACH, we collected only the successful trajectories, resulting in a success rate of $100\%$ for experts on these tasks. The expert performance for WALKER and CARRACING is $5637 \pm 55$ and $933 \pm 0.9$, respectively.

## C   Extended results of generalization experiments

### C.1   Experiment settings

To show our approach's better generalization capabilities, we extend the environment scenarios following the setting stated in Lee et al. [33]: (1) In MAZE main experiment, the initial and goal states of the expert dataset only constitute $50\%$ of the potential initial and goal states. In the generalization experiment, we gather expert demonstrations from some lower and higher coverage: $25\%$, $75\%$, and $100\%$. (2) In FETCHPICK, FETCHPUSH, and HANDROTATE main experiments, the demonstrations are collected in a lower noise level setting, $1\times$. Yet, the agent is trained within an environment incorporating $1.5\times$ noise, which is 1.5 times larger noise than the collected expert demonstration, applied to the starting and target block positions. In the generalization experiment, we train agents in different noise levels: $1\times$, $1.25\times$, $1.75\times$, $2.0\times$. (3) In ANTREACH main experiment, no random noise is added to the initial pose during policy learning. In ANTREACH generalization experiment, we train agents in different noise levels: 0 (default), 0.01, 0.03, 0.05.

These generalization experiments simulate real-world conditions. For example, because of the expenses of demonstration collection, the demonstrations may inadequately cover the entire state space, as seen in setup (1). Similarly, in setups (2) and (3), demonstrations may be acquired under controlled conditions with minimal noise, whereas the agent operating in a real environment would face more significant noise variations not reflected in the demonstrations, resulting in a broader distribution of initial states.

### C.2   Experiment results

**MAZE.** Our DRAIL outperforms baselines or performs competitively against DiffAIL across all demonstration coverages, as shown in Figure 8. Particularly, BC, WAIL, and GAIL's performance decline rapidly in the low coverage case. In contrast, diffusion model-based AIL algorithms demonstrate sustained performance, as shown in Figure 4a. This suggests that our method exhibits robust generalization, whereas BC and GAIL struggle with unseen states under limited demonstration coverage.

**FETCHPICK and FETCHPUSH.** In FETCHPICK, our method outperforms all baselines in most noise levels, as shown in Figure 8. In the $2.00\times$ noise level, our method DRAIL achieves a success rate of $87.22$, surpassing the best-performing baseline Diffusion Policy, which achieves only around $76.64\%$. GAIL, on the other hand, experiences failure in 3 out of the 5 seeds, resulting in a high standard deviation (mean: $39.17$, standard deviation: $47.98$) despite our thorough exploration of various settings for its configuration. In FETCHPUSH, our method DRAIL exhibits more robust results, in generalizing to unseen states compared to other baselines. This showcases that the diffusion

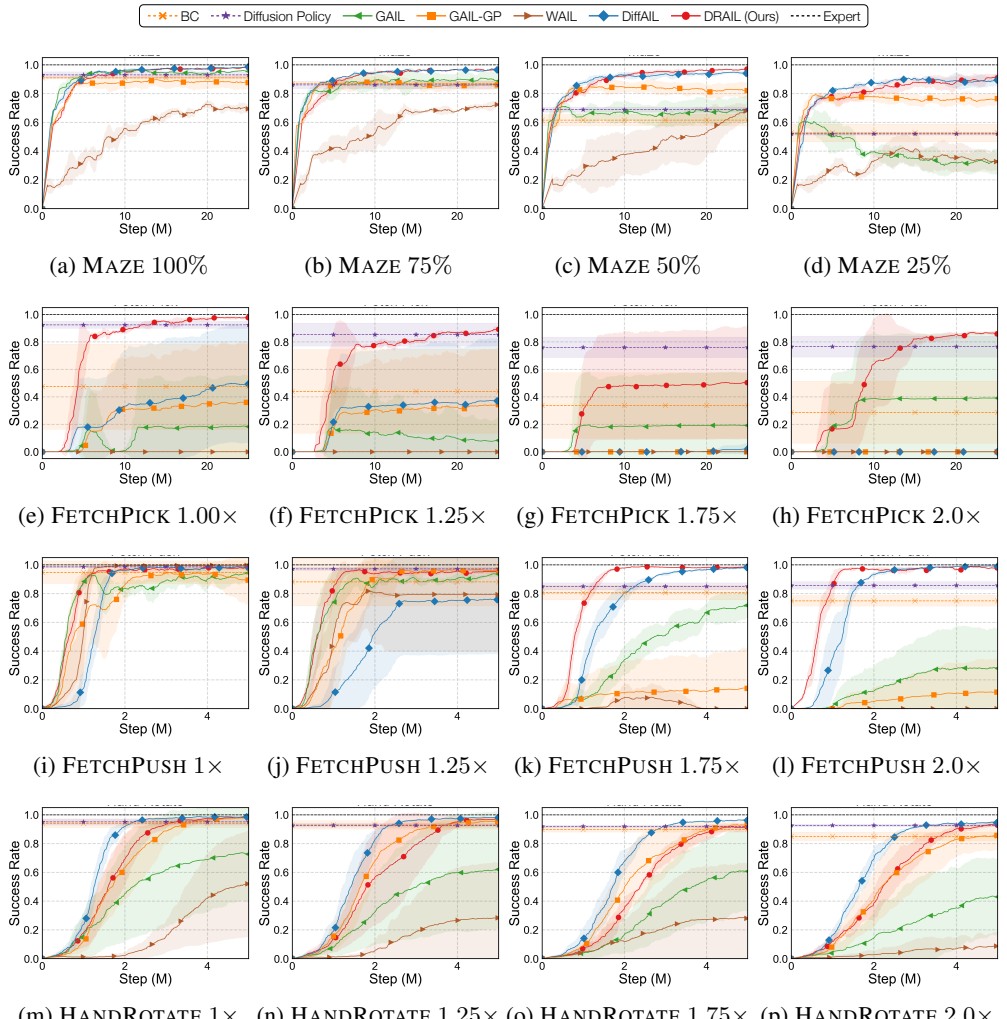

Figure 8: **Extended results of generalization experiments.** MAZE is evaluated with different coverages of state and goal locations in the expert demonstrations, while FETCHPICK, FETCHPUSH, and HANDROTATE environments are evaluated in environments of different noise levels. The number indicates the amount of additional noise in agent learning compared to that in the expert demonstrations, with more noise requiring harder generalization. The noise level rises from left to right.

reward guidance could provide better generalizability for the AIL framework. Moreover, our DRAIL is quite sample-efficient regarding interaction with the environment during training compared to other baselines in FETCHPICK and FETCHPUSH environment.

**HANDROTATE.** DiffAIL and Our DRAIL show robustness to different levels of noise in HANDRO-TATE, as illustrated in Figure 8. Specifically, DiffAIL and our DRAIL achieve a success rate of higher than 90% at a noise level of 2.0, while GAIL and WAIL only reach approximately 42.82% and 8.80%, respectively.

# D Hyperparameter Sensitivity Experiment

We empirically found that our proposed method, DRAIL, is robust to hyperparameters and easy to tune, especially compared to GAIL and WAIL. In this section, we present additional ablation experiments to examine how hyperparameter tuning affects the performance of DRAIL.

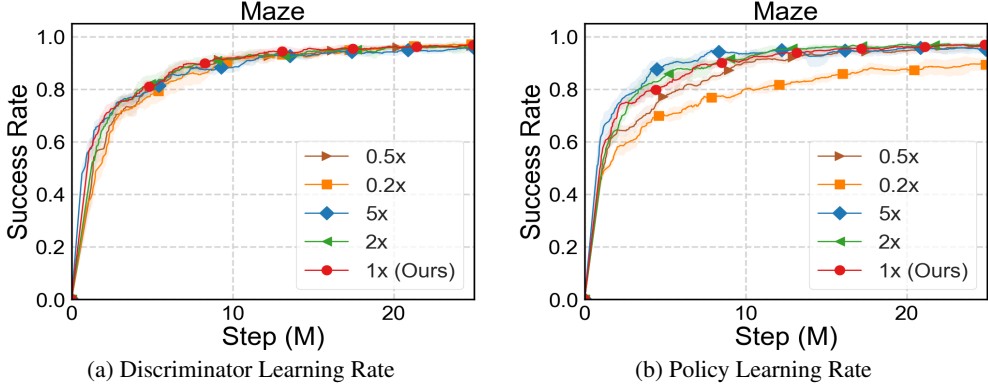

|  | Maze | | Maze |
|---|---|---|---|

(a) Discriminator Learning Rate            (b) Policy Learning Rate

Figure 9: **Hyperparameter Sensitivity of DRAIL in the MAZE Environment.** The results show the performance of DRAIL under varying learning rates for the discriminator (**a**) and policy (**b**). Different scaling factors (5x, 2x, 1x, 0.5x, 0.2x) of the baseline learning rate are tested. The results demonstrate that DRAIL remains robust across these variations, maintaining stable performance in the MAZE environment.

Like most AIL methods, the key hyperparameters of DRAIL are the learning rates of the policy and discriminator. the key hyperparameters for DRAIL are the learning rates of the policy and discriminator. We experimented with various learning rate values, including 5x, 2x, 1x, 0.5x, and 0.2x of the value used in the main results for the MAZE environment. The results, presented in Figure 9, demonstrate that our method is robust to variations in hyperparameters.

## E    Converged performance

This section reports the quantitative results of the converged performance across all experiments, including the main results in Section 5.3, the generalization experiments in Appendix C.2, and the data efficiency experiments in Section 5.5. The results are presented in Table 1.

## F    Model architecture

This section presents the model architecture of all the experiments. Appendix F.1 describe the model architecture of all methods used in Section 5.3.

### F.1    Model architecture of DRAIL, DiffAIL, and the baselines

In Section 5.3, we conducted a comparative analysis between our proposed DRAIL, along with several baseline approaches (BC, Diffusion Policy, GAIL, GAIL-GP, WAIL, and DiffAIL) across six diverse environments. We applied Multilayer Perceptron (MLP) for the policy of BC, the conditional diffusion model in Diffusion Policy, as well as the policy and the discriminator of GAIL, GAIL-GP, and WAIL. For DiffAIL and our proposed DRAIL, MLPs were employed in the policy and diffusion model of the diffusion discriminative classifier. The activation functions used for the MLPs in the diffusion model were ReLU, while hyperbolic tangent was employed for the others. The total timestep $T$ for all diffusion models in this paper is set to 1000 and the scheduler used for diffusion models is cosine scheduler [38]. Further details regarding the parameters for the model architecture can be found in Table 2.

**BC.**    We maintained a concise model for the policy of BC to prevent excessive overfitting to expert demonstrations. This precaution is taken to mitigate the potential adverse effects on performance when confronted with environments exhibiting higher levels of noise.

**Diffusion Policy.**    Based on empirical results and Chen et al. [5], the Diffusion Policy performs better when implemented with a deeper architecture. Consequently, we have chosen to set the policy's number of layers to 5.

Table 1: **Converged performance.** We report the quantitative results of the converged performance across all experiments.

| Environments | Settings | BC | Diffusion Policy | GAIL | WAIL | DiffAIL | DRAIL |
|---|---|---|---|---|---|---|---|
| | | | | **Main Results** | | | |
| MAZE | | $61.60\% \pm 3.98\%$ | $68.80\% \pm 1.72\%$ | $68.62\% \pm 8.35\%$ | $63.85\% \pm 5.77\%$ | $94.06\% \pm 1.67\%$ | $\mathbf{96.83\% \pm 0.49\%}$ |
| FETCHPUSH | | $82.00\% \pm 21.13\%$ | $89.40\% \pm 2.06\%$ | $33.73\% \pm 37.63\%$ | $91.64\% \pm 9.08\%$ | $\mathbf{93.66\% \pm 3.15\%}$ | $\mathbf{95.78\% \pm 0.80\%}$ |
| HANDROTATE | | $91.31\% \pm 2.61\%$ | $93.24\% \pm 1.61\%$ | $43.82\% \pm 35.31\%$ | $29.39\% \pm 35.65\%$ | $\mathbf{97.49\% \pm 0.48\%}$ | $93.88\% \pm 3.23\%$ |
| ANTREACH | | $33.00\% \pm 4.34\%$ | $33.60\% \pm 4.96\%$ | $21.98\% \pm 5.99\%$ | $33.84\% \pm 2.63\%$ | $29.59\% \pm 1.05\%$ | $\mathbf{39.02\% \pm 1.40\%}$ |
| WALKER | | $\mathbf{5431.19 \pm 267.40}$ | $4749.03 \pm 224.96$ | $3581.18 \pm 574.20$ | $2521.07 \pm 863.44$ | $4639.47 \pm 282.92$ | $5381.15 \pm 115.25$ |
| CARRACING | | $534.92 \pm 45.02$ | $592.20 \pm 103.17$ | $297.63 \pm 294.52$ | $-72.39 \pm 30.54$ | $689.62 \pm 96.72$ | $\mathbf{746.35 \pm 31.28}$ |
| | | | | **Generalization Experiments** | | | |
| MAZE | 25% | $52.67\% \pm 6.18\%$ | $52.00\% \pm 0.82\%$ | $33.05\% \pm 9.35\%$ | $32.53\% \pm 6.33\%$ | $89.67\% \pm 4.33\%$ | $\mathbf{91.00\% \pm 3.24\%}$ |
| | 50% | $61.60\% \pm 3.98\%$ | $68.80\% \pm 1.72\%$ | $68.62\% \pm 8.35\%$ | $63.85\% \pm 5.77\%$ | $94.06\% \pm 1.67\%$ | $\mathbf{96.83\% \pm 0.49\%}$ |
| | 75% | $87.00\% \pm 0.00\%$ | $86.00\% \pm 2.45\%$ | $88.87\% \pm 5.79\%$ | $72.47\% \pm 1.05\%$ | $\mathbf{96.58\% \pm 0.31\%}$ | $96.83\% \pm 0.58\%$ |
| | 100% | $91.00\% \pm 0.82\%$ | $93.03\% \pm 2.13\%$ | $95.47\% \pm 0.35\%$ | $69.52\% \pm 2.60\%$ | $\mathbf{98.43\% \pm 0.12\%}$ | $98.20\% \pm 0.04\%$ |
| FETCHPICK | 1.00× | $47.40\% \pm 31.10\%$ | $92.40\% \pm 2.58\%$ | $18.37\% \pm 36.74\%$ | $0.00\% \pm 0.00\%$ | $49.24\% \pm 42.53\%$ | $\mathbf{97.79\% \pm 1.27\%}$ |
| | 1.25× | $44.00\% \pm 30.63\%$ | $\mathbf{85.40\% \pm 8.40\%}$ | $8.57\% \pm 16.92\%$ | $0.00\% \pm 0.00\%$ | $37.03\% \pm 45.36\%$ | $89.22\% \pm 2.20\%$ |
| | 1.50× | $40.60\% \pm 27.62\%$ | $\mathbf{80.00\% \pm 7.13\%}$ | $19.03\% \pm 38.06\%$ | $0.00\% \pm 0.00\%$ | $33.97\% \pm 41.61\%$ | $86.90\% \pm 1.90\%$ |
| | 1.75× | $33.80\% \pm 23.96\%$ | $\mathbf{76.06\% \pm 7.48\%}$ | $19.15\% \pm 38.25\%$ | $0.00\% \pm 0.00\%$ | $2.14\% \pm 4.28\%$ | $49.86\% \pm 40.73\%$ |
| | 2.00× | $28.80\% \pm 22.68\%$ | $76.64\% \pm 7.63\%$ | $39.17\% \pm 47.98\%$ | $0.00\% \pm 0.00\%$ | $0.00\% \pm 0.00\%$ | $\mathbf{87.22\% \pm 1.93\%}$ |
| FETCHPUSH | 1.00× | $96.00\% \pm 6.10\%$ | $98.60\% \pm 0.49\%$ | $94.18\% \pm 4.87\%$ | $\mathbf{99.68\% \pm 0.30\%}$ | $98.12\% \pm 0.75\%$ | $97.90\% \pm 0.53\%$ |
| | 1.25× | $88.80\% \pm 17.07\%$ | $\mathbf{97.20\% \pm 0.98\%}$ | $94.34\% \pm 1.62\%$ | $79.32\% \pm 39.66\%$ | $75.66\% \pm 37.86\%$ | $95.84\% \pm 1.18\%$ |
| | 1.50× | $82.00\% \pm 21.13\%$ | $89.40\% \pm 2.06\%$ | $33.73\% \pm 37.63\%$ | $91.64\% \pm 9.08\%$ | $93.66\% \pm 3.15\%$ | $95.78\% \pm 0.80\%$ |
| | 1.75× | $71.00\% \pm 23.13\%$ | $84.80\% \pm 2.32\%$ | $43.71\% \pm 36.51\%$ | $0.00\% \pm 0.00\%$ | $55.62\% \pm 45.56\%$ | $\mathbf{96.15\% \pm 2.33\%}$ |
| | 2.00× | $64.40\% \pm 21.70\%$ | $79.20\% \pm 3.76\%$ | $16.89\% \pm 25.08\%$ | $0.00\% \pm 0.00\%$ | $40.90\% \pm 47.59\%$ | $\mathbf{99.09\% \pm 0.47\%}$ |
| HANDROTATE | 1.00× | $94.03\% \pm 2.90\%$ | $95.06\% \pm 2.16\%$ | $72.78\% \pm 33.57\%$ | $51.95\% \pm 36.88\%$ | $\mathbf{98.60\% \pm 0.16\%}$ | $98.32\% \pm 0.39\%$ |
| | 1.25× | $93.00\% \pm 2.94\%$ | $92.71\% \pm 1.27\%$ | $62.05\% \pm 41.21\%$ | $28.30\% \pm 38.68\%$ | $\mathbf{98.03\% \pm 0.61\%}$ | $97.17\% \pm 1.50\%$ |
| | 1.50× | $91.31\% \pm 2.61\%$ | $93.24\% \pm 1.61\%$ | $43.82\% \pm 35.31\%$ | $29.39\% \pm 35.65\%$ | $\mathbf{97.49\% \pm 0.48\%}$ | $93.88\% \pm 3.23\%$ |
| | 1.75× | $89.67\% \pm 1.70\%$ | $92.00\% \pm 0.82\%$ | $60.32\% \pm 27.33\%$ | $28.12\% \pm 37.87\%$ | $\mathbf{96.20\% \pm 0.25\%}$ | $91.73\% \pm 3.18\%$ |
| | 2.00× | $85.00\% \pm 2.94\%$ | $92.72\% \pm 0.91\%$ | $42.82\% \pm 26.72\%$ | $8.80\% \pm 9.77\%$ | $\mathbf{94.52\% \pm 0.80\%}$ | $92.77\% \pm 0.77\%$ |
| ANTREACH | 0.00× | $\mathbf{96.33\% \pm 0.94\%}$ | $95.33\% \pm 1.89\%$ | $76.80\% \pm 2.52\%$ | $75.67\% \pm 1.04\%$ | $71.35\% \pm 0.86\%$ | $83.03\% \pm 0.63\%$ |
| | 0.01× | $\mathbf{64.67\% \pm 5.25\%}$ | $63.33\% \pm 0.94\%$ | $13.75\% \pm 4.64\%$ | $28.03\% \pm 6.72\%$ | $26.83\% \pm 1.47\%$ | $39.50\% \pm 1.91\%$ |
| | 0.03× | $33.00\% \pm 4.34\%$ | $33.60\% \pm 4.96\%$ | $21.98\% \pm 5.99\%$ | $33.84\% \pm 2.63\%$ | $29.59\% \pm 1.05\%$ | $\mathbf{39.02\% \pm 1.40\%}$ |
| | 0.05× | $17.33\% \pm 2.87\%$ | $21.33\% \pm 0.94\%$ | $\mathbf{30.43\% \pm 1.03\%}$ | $21.90\% \pm 2.15\%$ | $13.58\% \pm 0.81\%$ | $29.92\% \pm 3.74\%$ |
| | | | | **Data Efficiency** | | | |
| FETCHPUSH | 20311 | $82.00\% \pm 21.13\%$ | $89.40\% \pm 2.06\%$ | $33.73\% \pm 37.63\%$ | $91.64\% \pm 9.08\%$ | $\mathbf{93.66\% \pm 3.15\%}$ | $\mathbf{95.78\% \pm 0.80\%}$ |
| | 10000 | $80.20\% \pm 14.70\%$ | $90.20\% \pm 2.64\%$ | $24.18\% \pm 31.75\%$ | $62.10\% \pm 43.86\%$ | $55.79\% \pm 45.62\%$ | $\mathbf{95.31\% \pm 0.45\%}$ |
| | 5000 | $76.40\% \pm 26.27\%$ | $86.60\% \pm 2.15\%$ | $43.59\% \pm 36.58\%$ | $77.60\% \pm 38.91\%$ | $38.04\% \pm 46.32\%$ | $\mathbf{95.70\% \pm 0.75\%}$ |
| | 2000 | $\mathbf{80.20\% \pm 14.70\%}$ | $\mathbf{84.40\% \pm 1.85\%}$ | $33.28\% \pm 24.07\%$ | $41.21\% \pm 47.47\%$ | $46.61\% \pm 37.37\%$ | $86.44\% \pm 13.44\%$ |
| WALKER | 5 trajs | $\mathbf{5431.15 \pm 267.40}$ | $4749.03 \pm 224.96$ | $3581.18 \pm 574.20$ | $2521.07 \pm 863.44$ | $4639.47 \pm 282.92$ | $5381.15 \pm 115.25$ |
| | 3 trajs | $5061.72 \pm 73.57$ | $3476.26 \pm 313.01$ | $2837.76 \pm 1028.95$ | $3210.65 \pm 518.24$ | $4584.24 \pm 200.69$ | $\mathbf{5266.41 \pm 57.93}$ |
| | 2 trajs | $\mathbf{4957.94 \pm 658.54}$ | $1872.29 \pm 325.88$ | $2323.08 \pm 886.06$ | $2067.75 \pm 409.50$ | $3250.23 \pm 1610.07$ | $5083.32 \pm 119.98$ |
| | 1 traj | $3055.27 \pm 1834.75$ | $1165.29 \pm 147.79$ | $960.65 \pm 79.89$ | $1017.37 \pm 47.21$ | $3057.35 \pm 1530.88$ | $\mathbf{4960.10 \pm 164.57}$ |

**GAIL & GAIL-GP.** The detailed model architecture for GAIL and GAIL-GP is presented in Table 2. For GAIL-GP, the gradient penalty is set to 1 across all environments.

**WAIL.** We set the ground transport cost and the type of regularization of WAIL as Euclidean distance and $L_2$-regularization. The regularization value $\epsilon$ is provided in Table 2.

**DiffAIL.** In DiffAIL, the conditional diffusion model is not utilized as it only needs to consider the numerator of Equation (4). Consequently, the diffusion model takes only the noisy state-action pairs as input and outputs the predicted noise value.

**DRAIL.** The conditional diffusion model of the diffusion discriminative classifier in our DRAIL is constructed by concatenating either the real label $\mathbf{c}^+$ or the fake label $\mathbf{c}^-$ to the noisy state-action pairs as the input. The model then outputs the predicted noise applied to the state-action pairs. The dimensions of both $\mathbf{c}^+$ and $\mathbf{c}^-$ are reported in Table 2.

### F.2 Image-based model architecture of DRAIL, DiffAIL, and the baselines

For the CARRACING task, we redesigned the model architecture for our DRAIL and all baseline methods to handle image-based input effectively.

In CARRACING, the policy for all baselines utilizes a convolutional neural network (CNN) for feature extraction followed by a multi-layer perceptron (MLP) for action prediction. The CNN consists of three downsampling blocks with 32, 64, and 64 channels respectively. The kernel sizes for these blocks are 8, 4, and 3, with strides of 4, 2, and 1, respectively. After feature extraction, the output is flattened and passed through a linear layer to form a 512-dimensional feature vector representing the state data. This state feature vector is subsequently processed by an MLP with three layers, each

Table 2: **Model architectures of policies and discriminators.** We report the architectures used for all the methods on all the tasks. Note that $\pi$ denotes the neural network policy, $D$ represents a multilayer perceptron discriminator used in GAIL, GAIL-GP, and WAIL, and $D_\phi$ represents a diffusion model discriminator used in DiffAIL and our method DRAIL.

| Method | Models | Component | MAZE | FETCHPICK | FETCHPUSH | HANDROTATE | ANTREACH | WALKER |
|---|---|---|---|---|---|---|---|---|
| BC | $\pi$ | # Layers | 3 | 4 | 3 | 4 | 3 | 3 |
| | | Input Dim. | 6 | 16 | 16 | 68 | 132 | 17 |
| | | Hidden Dim. | 256 | 256 | 256 | 512 | 256 | 256 |
| | | Output Dim. | 2 | 4 | 3 | 20 | 8 | 6 |
| Diffusion Policy | $\pi$ | # Layers | 5 | 5 | 5 | 5 | 6 | 7 |
| | | Input Dim. | 8 | 20 | 19 | 88 | 140 | 23 |
| | | Hidden Dim. | 256 | 1200 | 1200 | 2100 | 1200 | 1024 |
| | | Output Dim. | 2 | 4 | 3 | 20 | 8 | 6 |
| GAIL & GAIL-GP | $D$ | # Layers | 3 | 4 | 5 | 4 | 5 | 5 |
| | | Input Dim. | 8 | 20 | 19 | 88 | 140 | 23 |
| | | Hidden Dim. | 64 | 64 | 64 | 128 | 64 | 64 |
| | | Output Dim. | 1 | 1 | 1 | 1 | 1 | 1 |
| | $\pi$ | # Layers | 3 | 3 | 3 | 3 | 3 | 3 |
| | | Input Dim. | 6 | 16 | 16 | 68 | 132 | 17 |
| | | Hidden Dim. | 64 | 64 | 256 | 64 | 256 | 256 |
| | | Output Dim. | 2 | 4 | 3 | 20 | 8 | 6 |
| WAIL | $D$ | # Layers | 3 | 4 | 5 | 4 | 5 | 5 |
| | | Input Dim. | 8 | 20 | 19 | 88 | 140 | 23 |
| | | Hidden Dim. | 64 | 64 | 64 | 128 | 64 | 64 |
| | | Output Dim. | 1 | 1 | 1 | 1 | 1 | 1 |
| | | Reg. Value $\epsilon$ | 0 | 0 | 0.01 | 0 | 0.01 | 0.1 |
| | $\pi$ | # Layers | 3 | 3 | 3 | 3 | 3 | 3 |
| | | Input Dim. | 6 | 16 | 16 | 68 | 132 | 17 |
| | | Hidden Dim. | 64 | 64 | 256 | 64 | 256 | 256 |
| | | Output Dim. | 2 | 4 | 3 | 20 | 8 | 6 |
| DiffAIL | $D_\phi$ | # Layers | 5 | 4 | 5 | 3 | 5 | 5 |
| | | Input Dim. | 8 | 20 | 19 | 88 | 140 | 23 |
| | | Hidden Dim. | 128 | 128 | 1024 | 128 | 1024 | 1024 |
| | | Output Dim. | 8 | 20 | 19 | 88 | 140 | 23 |
| | $\pi$ | # Layers | 3 | 3 | 3 | 3 | 3 | 3 |
| | | Input Dim. | 6 | 16 | 16 | 68 | 132 | 17 |
| | | Hidden Dim. | 64 | 64 | 256 | 64 | 256 | 256 |
| | | Output Dim. | 2 | 4 | 3 | 20 | 8 | 6 |
| DRAIL (Ours) | $D_\phi$ | # Layers | 5 | 4 | 5 | 3 | 5 | 5 |
| | | Input Dim. | 18 | 30 | 29 | 98 | 150 | 33 |
| | | Hidden Dim. | 128 | 128 | 1024 | 128 | 1024 | 1024 |
| | | Output Dim. | 8 | 20 | 19 | 88 | 140 | 23 |
| | | Label Dim. $|\mathbf{c}|$ | 10 | 10 | 10 | 10 | 10 | 10 |
| | $\pi$ | # Layers | 3 | 3 | 3 | 3 | 3 | 3 |
| | | Input Dim. | 6 | 16 | 16 | 68 | 132 | 17 |
| | | Hidden Dim. | 64 | 64 | 256 | 64 | 256 | 256 |
| | | Output Dim. | 2 | 4 | 3 | 20 | 8 | 6 |

having a hidden dimension of 256, to predict the appropriate action. In Diffusion Policy, we only use the downsampling part to extract features.

**Diffusion Policy.** Diffusion Policy represents a policy as a conditional diffusion model, which predicts an action conditioning on a state and a randomly sampled noise. Our condition diffusion model is implemented using the diffusers package by von Platen et al. [57]. The state in CARRACING image of size $64 \times 64$, so we first use a convolutional neural network (CNN) to extract the feature. The CNN is based on a U-Net [47] structure, comprising 3 down-sampling blocks. Each block consists of 2 ResNet [19] layers, with group normalization applied using 4 groups. The channel sizes for each pair of down-sampling blocks are 4, 8, and 16, respectively.

**Discriminator of GAIL, GAIL-GP, & WAIL.** The discriminators of GAIL, GAIL-GP, and WAIL are similar to the policy model; the only difference is that the last linear layer outputs a 1-dimensional value that indicates the probability of a given state-action pair being from the expert demonstrations.

**Discriminator of DiffAIL & DRAIL.** Our diffusion model is implemented using the diffusers package by von Platen et al. [57]. The architecture of both DiffAIL and DRAIL is based on a U-Net

Table 3: **Hyperparameters.** This table provides an overview of the hyperparameters used for all methods across various tasks. $\eta_\phi$ denotes the learning rate of the discriminator, while $\eta_\pi$ denotes the learning rate of the policy.

| Method | Hyperparameter | MAZE | FETCHPICK | FETCHPUSH | HANDROTATE | ANTREACH | WALKER | CARRACING |
|---|---|---|---|---|---|---|---|---|
| BC | Learning Rate | 0.00005 | 0.0008 | 0.0002 | 0.0001 | 0.001 | 0.0001 | 0.0003 |
|  | Batch Size | 128 | 128 | 128 | 128 | 128 | 128 | 128 |
|  | # Epochs | 2000 | 1000 | 1000 | 5000 | 1000 | 1000 | 25000 |
| Diffusion Policy | Learning Rate | 0.0002 | 0.00001 | 0.0001 | 0.0001 | 0.00001 | 0.0001 | 0.0001 |
|  | Batch Size | 128 | 128 | 128 | 128 | 128 | 128 | 128 |
|  | # Epochs | 20000 | 20000 | 10000 | 2000 | 10000 | 5000 | 100000 |
| GAIL & GAIL-GP | $\eta_\phi$ | 0.001 | 0.00001 | 0.000008 | 0.0001 | 0.0001 | 0.0000005 | 0.0001 |
|  | $\eta_\pi$ | 0.0001 | 0.00005 | 0.0002 | 0.0001 | 0.0001 | 0.0001 | 0.0001 |
|  | Env. Steps | 25000000 | 25000000 | 5000000 | 5000000 | 10000000 | 25000000 | 2000000 |
| WAIL | $\eta_\phi$ | 0.00001 | 0.0001 | 0.00008 | 0.0001 | 0.00001 | 0.0000008 | 0.0001 |
|  | $\eta_\pi$ | 0.00001 | 0.0005 | 0.0001 | 0.0001 | 0.0001 | 0.0001 | 0.0001 |
|  | Env. Steps | 25000000 | 25000000 | 5000000 | 5000000 | 10000000 | 25000000 | 2000000 |
| DiffAIL | $\eta_\phi$ | 0.001 | 0.0001 | 0.0001 | 0.0001 | 0.0001 | 0.0001 | 0.00001 |
|  | $\eta_\pi$ | 0.0001 | 0.0001 | 0.00005 | 0.0001 | 0.0001 | 0.0001 | 0.0001 |
|  | Env. Steps | 25000000 | 25000000 | 5000000 | 5000000 | 10000000 | 25000000 | 20000000 |
| DRAIL (Ours) | $\eta_\phi$ | 0.001 | 0.0001 | 0.001 | 0.0001 | 0.001 | 0.0002 | 0.0001 |
|  | $\eta_\pi$ | 0.0001 | 0.0001 | 0.0001 | 0.0001 | 0.0001 | 0.0001 | 0.0001 |
|  | Env. Steps | 25000000 | 25000000 | 5000000 | 5000000 | 10000000 | 25000000 | 20000000 |

Table 4: **PPO training parameters.** This table reports the PPO training hyperparameters used for each task.

| Hyperparameter | MAZE | FETCHPICK | FETCHPUSH | HANDROTATE | ANTREACH | WALKER | CARRACING |
|---|---|---|---|---|---|---|---|
| Clipping Range $\epsilon$ | 0.2 | 0.2 | 0.2 | 0.2 | 0.2 | 0.2 | 0.2 |
| Discount Factor $\gamma$ | 0.99 | 0.99 | 0.99 | 0.99 | 0.99 | 0.99 | 0.99 |
| GAE Parameter $\lambda$ | 0.95 | 0.95 | 0.95 | 0.95 | 0.95 | 0.95 | 0.95 |
| Value Function Coefficient | 0.5 | 0.5 | 0.5 | 0.5 | 0.5 | 0.5 | 0.5 |
| Entropy Coefficient | 0.0001 | 0.0001 | 0.001 | 0.0001 | 0.001 | 0.001 | 0 |

[47] structure, comprising 3 down-sampling blocks and 3 up-sampling blocks. Each block consists of 2 ResNet [19] layers, with group normalization applied using 4 groups. The channel sizes for each pair of down-sampling and up-sampling blocks are 4, 8, and 16, respectively. The condition label is incorporated through class embedding, with the number of classes set to 2, representing the real label $\mathbf{c}^+$ and the fake label $\mathbf{c}^-$. Finally, we apply running normalization at the output to ensure stable training and accurate discrimination.

To accommodate both image-based state and vector-based action data within the diffusion model, we flatten the action data into an image with a channel size equivalent to the action dimension. Subsequently, we concatenate the state and transformed action data as input to the U-Net. In DRAIL, we use 1 to represent $\mathbf{c}^+$ and 0 to represent $\mathbf{c}^-$ as condition data. For DiffAIL, since condition labels are not required, we simply assign a constant value of 0 as the condition label.

## G    Training details

### G.1    Training hyperparamters

The hyperparameters employed for all methods across various tasks are outlined in Table 3. The Adam optimizer [27] is utilized for all methods, with the exception of the discriminator in WAIL, for which RMSProp is employed. Linear learning rate decay is applied to all policy models.

Due to the potential impact of changing noise levels on the quality of agent data input for the discriminator, the delicate balance between the discriminator and the AIL method's policy may be disrupted. Therefore, we slightly adjusted the learning rate for the policy and the discriminator for different noise levels on each task. The reported parameters in Table 3 correspond to the noise levels presented in Figure 4.

### G.2    Reward function details

As explained in Section 4.3, we adopt the optimization objective proposed by [12] as diffusion reward signal for the policy learning in our DRAIL. To maintain fairness in comparisons, we apply the same reward function to DiffAIL and GAIL. In CARRACING, we observe that adapting GAIL's optimization objective could lead to better performance; hence, we use it for DRAIL, DiffAIL, and

GAIL. For WAIL, we adhere to the approach outlined in the original paper, wherein the output of the discriminator directly serves as the reward function.

In our experiments, we employ Proximal Policy Optimization (PPO) [50], a widely used policy optimization method, to optimize policies for all the AIL methods. We maintain all hyperparameters of PPO constant across methods for a given task, except the learning rate, which is adjusted for each method. The PPO hyperparameters for each task are presented in Table 4.

## H  Limitations

This work presents an adversarial imitation learning framework DRAIL by employing a diffusion model as a discriminator. While DRAIL achieves encouraging results in various domains, including robot arm manipulation, robot hand dexterous manipulation, locomotion, and games, our proposed framework is fundamentally limited to the learning from demonstration (LfD) setting. That said, DRAIL requires both state and action sequences and, therefore, cannot learn from videos or state-only sequences, *i.e.*, learning from observation (LfO). Moreover, DRAIL assumes expert demonstrations to be optimal, and its performance may not be satisfactory if the demonstrations contain a certain level of noise or the demonstrators are suboptimal. Finally, DRAIL with its imitation learning nature, is not designed to learn from environmental rewards; therefore, even when environments can provide rewards, there is no apparent mechanism to utilize them with the current formulation of DRAIL.

## I  Computational resources and time

### I.1  Computational resources

For our experiments, we used the following three workstations:

- Machine 1 & Machine 2: ASUS WS880T workstation
  - CPU: an Intel Xeon W-2255 (10C/20T, 19.25M, 4.5GHz) 48-Lane CPU
  - GPUs: an NVIDIA RTX 3080 Ti GPU and an NVIDIA RTX 3090 GPU
  - Memory: 128GB memory
- Machine 3: ASUS WS880T workstation
  - CPU: an Intel Xeon W-2255 (10C/20T, 19.25M, 4.5GHz) 48-Lane CPU
  - GPUs: two NVIDIA RTX 3080 Ti GPUs
  - Memory: 128GB memory

### I.2  Computational time

In the following, we report the total approximate training GPU hours for all algorithms across all environments, with each algorithm trained on 5 random seeds.

- Main Experiments: 1945 GPU hours
- Generalization Experiments: 7300 GPU hours
- Data Efficiency Experiments: 2920 GPU hours
- Reward Function Visualization Experiments: 8 GPU hours

We conducted all the experiments on the following three workstations:

- M1: ASUS WS880T workstation with an Intel Xeon W-2255 (10C/20T, 19.25M, 4.5GHz) 48-Lane CPU, 64GB memory, an NVIDIA RTX 3080 Ti GPU, and an NVIDIA RTX 3090 Ti GPU
- M2: ASUS WS880T workstation with an Intel Xeon W-2255 (10C/20T, 19.25M, 4.5GHz) 48-Lane CPU, 64GB memory, an NVIDIA RTX 3080 Ti GPU, and an NVIDIA RTX 3090 Ti GPU
- M3: ASUS WS880T workstation with an Intel Xeon W-2255 (10C/20T, 19.25M, 4.5GHz) 48-Lane CPU, 64GB memory, and two NVIDIA RTX 3080 Ti GPUs

## J  Impact statements

In this work, we propose a novel adversarial imitation learning framework, diffusion rewards guided adversarial imitation learning (DRAIL), which integrates a diffusion model into GAIL. The proposed framework can potentially reinforce the biases captured by expert demonstrations, which can lead to sub-optimal, unsafe, or even discriminatory behaviors. To address this issue, we encourage future works to focus on alleviating these issues in imitation learning, *e.g.*, fairness in machine learning, and responsible AI.

