# OpenReview forum: "Diffusion-Reward Adversarial Imitation Learning"
_NeurIPS.cc/2024/Conference — NeurIPS 2024 poster_

### Official Review · Reviewer_5b3J · 2024-07-02

**Soundness:** 3
**Presentation:** 2
**Contribution:** 3
**Rating:** 6
**Confidence:** 4

**Summary:**

The paper proposes Diffusion-Reward Adversarial Imitation Learning (DRAIL) an Adversarial Imitation Learning method where the discriminator is parameterized by the loss of a conditional diffusion model.

The diffusion model takes in a state-action pair $(s, a)$ and a binary label $c$ indicating whether the state-action pair comes from the discriminator (label $c^-$) or from the training set (label $c^+$). Its loss on these inputs is denoted $\mathcal{L}\_{\textrm{diff}}(s, a, c)$. The discriminator is then defined as $D(s, a) = \sigma(\mathcal{L}\_{\textrm{diff}}(s, a, c^+) - \mathcal{L}\_{\textrm{diff}}(s, a, c^-))$, where $\sigma$ is the sigmoid function.

The method resembles DiffAIL (Wang et al. 2023), which defines the discriminator as $\exp(-\mathcal{L}\_{\textrm{diff}}(s, a))$ (unconditional diffusion model), as well as Diffusion Classifier (Li et al. 2023).

They benchmark DRAIL on six settings, including long-horizon planning (e.g. Maze2d), high-dimensional control (e.g. AntReach) and dexterous manipulation (e.g. HandRotate). They compare against methods including Behavior Cloning, the original GAIL, and variants such as Wasserstein AIL and DiffAIL. They also compare against Diffusion Policy, which uses a diffusion model to parameterize a policy rather than a reward function.

In terms of raw performance, DRAIL obtains:

- visibly better results than the baselines in 4/6 environments;

- a result very close to that of DiffAIL in 1/6  environment;

- a result worse than DiffAIL in 1/6 environment.

Further experiments show:

- DRAIL is more robust to noise on initial states and goal positions than the baselines on a block-pushing task (FetchPush);

- DRAIL is more data-efficient than the baselines on the Walker environment (continuous high-dimensional control) and FetchPush;

- DRAIL learns more generalizable rewards than GAIL on a synthetic 2d point dataset.

**Strengths:**

1. Clarity: the paper is well-written and the proposed method is explained clearly.

1. Extensive comparison with DiffAIL, albeit in the appendix.

1. Methodological novelty: the authors propose using a conditional diffusion model to parameterize the discriminator for AIL, which removes limitations from DiffAIL (such as an arbitrary decision boundary at $\ln 2$; see Appendix A).

1. Significance and quality of experimental results: the paper demonstrates that defining the discriminator using a conditional diffusion model improves performance, robustness and data-efficiency, compared to DiffAIL (which uses an unconditional diffusion model in the discriminator) and to other baselines. Hence, one can argue it constitutes a step forward in Adversarial Imitation Learning.

1. Referencing strongly related prior work: the authors point to DiffAIL and Diffusion Classifier, which strongly resemble different parts of DRAIL (namely using the loss of a diffusion model to parameterize a discriminator, and how to construct such a discriminator). I personally was not aware of these two papers before, and commend the authors for bringing these to the reader’s attention.

**Weaknesses:**

## Better delimiting the author’s contributions

Despite DiffAIL and Diffusion Classifier being mentioned and their relationship to DRAIL being discussed, my impression is that more credit should be materially given to these two prior works.

For example, in line 143, the authors state “we propose to leverage the training procedure of DDPM and develop a mechanism to provide learning signals to a policy using a single denoising step”. However, it seems to me that this much was already present in DiffAIL.

Similarly, line 145 states “Our key insight is to leverage the derivations developed by Kingma et al. [22], Song et al. [47], which suggest that the diffusion loss, i.e., the difference between the predicted noise and the injected noise, indicates how well the data fits the target distribution since the diffusion loss is the upper bound of the negative log-likelihood of data in the target distribution”. However, it seems that a similar insight had already been used by Li et al. (2023), as mentioned in line 177.

Hence, my understanding is that, in the context of these two prior works, DRAIL can be seen as a combination of DiffAIL and Diffusion Classifier (DC). Please feel free to correct me if this understanding is incorrect.

This still constitutes novelty, although less than if DiffAIL and DC hadn’t preceded this work. In fact, in Appendix A the authors make what I consider to be a compelling case for why the discriminator should be “symmetric”, rather than “one-sided” like in DiffAIL.

As such, I believe this paper’s contributions could be presented in a way that better separates its original contribution from that present in prior work. One way of doing this could be:
- Add a section on DiffAIL and Diffusion Classifier to Section 3 (preliminaries).
- In Section 4 (Approach), discuss the limitations of DiffAIL, e.g. as done in Appendix A. Argue that using a conditional diffusion model with a symmetric discriminator (via Diffusion Classifier) would address these limitations.
- Hence present the final methodology as the combination of DiffAIL and DC.

Again, I am open to changing my mind about the above assessment, and will raise my score accordingly in that case, or in case a refactoring of the presentation of the method is implemented by the authors.

## Baselines against planning methods that use diffusion models

The authors baseline DRAIL against 3 AIL methods and Diffusion Policy. The motivation for including the latter is to “compare learning a diffusion model as a policy (diffusion policy) or reward function (ours)”. As per a recent survey of Zhu et al. (2023), diffusion models have been used in sequential decision-making not only for parameterizing policies, but also as planners, as is the case in the seminal work of Janner et al. (2022).

As such, for a more holistic comparison against methods outside of the AIL framework, the authors could include diffusion-based planning baselines such as Diffuser (Janner et al. 2022).

Recent work by Nuti et al. (2023) has also shown reward functions can be recovered from pairs of diffusion models in the setting of Janner et al. (2022), so that it would also be possible to compare the rewards learned via DRAIL against those obtained from Diffuser.


### References

Zhu, Zhengbang, et al. "Diffusion models for reinforcement learning: A survey." arXiv preprint arXiv:2311.01223 (2023).

Nuti, Felipe, Tim Franzmeyer, and João F. Henriques. "Extracting reward functions from diffusion models." Advances in Neural Information Processing Systems 36 (2024).

**Questions:**

See Weaknesses.

**Limitations:**

The authors consider the limitations of their work in Appendix G and societal impacts in Appendix I.

My main concern with this work is the extent to which the authors claim originality over aspects of their method that are arguably already present in prior work. If the authors can clarify that this concern is misplaced, or if they modify the exposition of their contributions as outlined in the Weaknesses sections, I am open to raising my score.

---

> ### Author Rebuttal · Authors · 2024-08-07
>
> We sincerely thank the reviewer for the thorough and constructive comments. Please find the response to your questions below.
>
> > Hence, my understanding is that, in the context of these two prior works, DRAIL can be seen as a combination of DiffAIL and Diffusion Classifier (DC). Please feel free to correct me if this understanding is incorrect.
>
> > This still constitutes novelty, ... In fact, in Appendix A the authors make what I consider to be a compelling case for why the discriminator should be “symmetric”, rather than “one-sided” like in DiffAIL. As such, I believe this paper’s contributions could be presented in a way that better separates its original contribution from that present in prior work.
>
> We thank the reviewer for the detailed suggestions for better organizing the paper and discussing the novelty of our work. We would like to clarify that DRAIL is not a combination of DiffAIL and Diffusion Classifier (DC).
>
> DC turns a text-to-image diffusion model, which optimizes the denoising MSE loss, into an image classifier without further training. DiffAIL follows the intuition of DC and uses how well a diffusion model can denoise a state-action pair to indicate the “realness” of the pair for adversarial imitation learning. To this end, DiffAIL learns an unconditional diffusion model to denoise expert state-action pairs well, while denoise agent state-action pairs poorly.
>
> In contrast, we propose to directly train an expert/agent binary classifier by optimizing the binary cross-entropy (BCE) loss according to the denoising performance of a conditional diffusion model conditioning on expert/agent label, and our novelty lies in such a design. In other words, our method significantly differs from DC and DiffAIL since we formulate distinguishing expert and agent state-action pairs as a binary classification task instead of a denoising task, and this design aligns with the GAIL formulation. Hence, our method is not a combination of DC and DiffAIL.
>
> Moreover, DiffAIL’s evaluation only considers locomotion tasks, while our work extensively compares our method in various domains, including navigation (Maze and AntReach), locomotion (Walker and AntReach), robot arm manipulation (FetchPush and FetchPick), robot arm dexterous (HandRotate), and games (CarRacing). Additionally, we present experimental results on generalization to unseen states and goals and on varying amounts of expert data.
>
> We completely agree with and highly appreciate the reviewer’s detailed suggestion for reorganizing the paper. We will revise the paper by presenting DC and DiffAIL in Section 3 and discussing the limitations of DiffAIL in Section 4 to motivate our method. We believe this will make our contributions more clear.
>
> > The authors could include diffusion-based planning baselines such as Diffuser (Janner et al. 2022). Recent work by Nuti et al. (2023) has also shown reward functions can be recovered from pairs of diffusion models in the setting of Janner et al. (2022),
>
> We thank the reviewer for providing these references. We will revise the paper to include the following discussions.
>
> - Diffuser [1] is a model-based RL method that requires trajectory-level reward information, which differs from our setting, i.e., imitation learning, where obtaining rewards is not possible. Therefore, it is not trivial to directly compare our method to the diffuser.
> - Nuti et al. [2] focus on learning a reward function, unlike imitation learning, whose goal is to obtain a policy. Hence, Nuti et al. [2] neither present policy learning results in the main paper nor compare their method to imitation learning methods. Moreover, they focus on learning from a fixed suboptimal dataset, while adversarial imitation learning (AIL) approaches and our method are designed to learn from agent data that continually change as the agents learn.
>
> **References**
>
> [1] Janner et al. “Planning with diffusion for flexible behavior synthesis.” In ICML, 2022.
>
> [2] Nuti et al. “Extracting Reward Functions from Diffusion Models.” In NeurIPS, 2024.

---

> > ### Author Response · Authors · 2024-08-10
> > **Reminder: The reviewer-author discussion period ends in three days**
> >
> > We would like to express our sincere gratitude to the reviewer for the thorough and constructive feedback. We are confident that our responses adequately address the concerns raised by the reviewer, including the following points.
> > - A clarification of our contributions and a detailed plan to reorganize our paper based on the reviewer's suggestion
> > - Discussions of Janner et al., 2022 (Diffuser) and Nuti et al., 2023 (Extracting Reward Functions from Diffusion Models)
> >
> > Please kindly let us know if the reviewer has any additional concerns or if further experimental results are required. We are fully committed to resolving any potential issues, should time permit. Again, we thank the reviewer for all the detailed review and the time the reviewer put into helping us to improve our submission.

---

### Official Review · Reviewer_PZWW · 2024-07-11

**Soundness:** 3
**Presentation:** 2
**Contribution:** 3
**Rating:** 5
**Confidence:** 4

**Summary:**

This paper aims to address the training instability problem in generative adversarial imitation learning. The authors propose a diffusion discriminative classifier that helps achieve a more stable policy learning process by enhancing the smoothness and robustness of the reward model. Additionally, the author experimentally validates the effectiveness of DRIL across various control tasks, including manipulation, locomotion, and navigation benchmarks.

**Strengths:**

- The instability of AIL training is a key problem to its application, making research on this issue valuable.
- DRAIL performs well across multiple benchmarks, improving sample efficiency and training stability.

**Weaknesses:**

- The advantages of the diffusion discriminative classifier compared to diffusion reward [1] are not well explained. The authors claim that their proposed diffusion discriminative classifier only requires two reverse steps to obtain the corresponding reward, thereby reducing the computational resource consumption needed for sampling in diffusion models. However, I do not fully agree with this. According to equations (4) and (6), DRAIL needs to compute $L_{\text{diff}}(s,a,c^+)$ and $L_{\text{diff}}(s,a,c^-)$ when obtaining rewards. From equation (3), we know both terms require taking an expectation over the diffusion steps $T$. Therefore, to obtain an accurate reward, a number of inference steps equivalent to the diffusion steps are still needed.
- This paper lacks ablation experiments, including a sensitivity analysis of the hyperparameters.
- Some important AIL baselines are missing, such as DAC [2] and IQLearn [3].

[1] DiffAIL: Diffusion Adversarial Imitation Learning. AAAI, 2024.

[2] Discriminator-Actor-Critic: Addressing Sample Inefficiency and Reward Bias in Adversarial Imitation Learning. ICLR, 2019.

[3] IQ-Learn: Inverse soft-Q Learning for Imitation. NeurIPS, 2021.

**Questions:**

- Why does DRAIL only require two reverse steps to obtain an accurate reward? Can you provide a more detailed explanation or analysis?
- I believe that the requirement for stable algorithm training includes robustness to hyperparameters. A major issue with AIL methods is the need for extensive hyperparameter tuning, and well-tuned AIL can achieve good performance. What are the key hyperparameters for DRAIL? Can you conduct additional ablation experiments on these hyperparameters?
- Can you compare DAC and IQLearn in your experiments? Both DAC and IQLearn use the gradient penalty for reward smoothing, and I am very interested in seeing a comparison between the diffusion discriminative classifier and the gradient penalty.

I would like to raise my score if my concerns are well addressed.

**Limitations:**

The authors have discussed the limitations of this paper. DRAIL is unable to learn from state-only trajectories and suboptimal trajectories.

---

> ### Author Rebuttal · Authors · 2024-08-07
>
> We sincerely thank the reviewer for the thorough and constructive comments. Please find the response to your questions below.
>
> > The advantages of the diffusion discriminative classifier compared to diffusion reward [1] are not well explained.
>
> We extensively discuss how our proposed method differs from DiffAIL and why our proposed method produces better rewards in Section A. We will reorganize the paper to make this clear from the main paper.
>
> DiffAIL uses how well a diffusion model can denoise a state-action pair to indicate the “realness” of the pair for adversarial imitation learning. To this end, DiffAIL learns an unconditional diffusion model to denoise expert state-action pairs well, while denoise agent state-action pairs poorly. In contrast, we propose to directly train an expert/agent binary classifier by optimizing the binary cross-entropy (BCE) loss according to the denoising performance of a conditional diffusion model conditioning on expert/agent label, and our novelty lies in such a design. Our method significantly differs from DiffAIL since we formulate distinguishing expert and agent state-action pairs as a binary classification task instead of a denoising task, and this design aligns better with the GAIL formulation compared to DiffAIL.
>
> Moreover, DiffAIL’s evaluation only considers locomotion tasks, while our work extensively compares our method in various domains, including navigation (Maze and AntReach), locomotion (Walker and AntReach), robot arm manipulation (FetchPush and FetchPick), robot arm dexterous (HandRotate), and games (CarRacing). Additionally, we present experimental results on generalization to unseen states and goals and on varying amounts of expert data.
>
> > The authors claim that their proposed diffusion discriminative classifier only requires two reverse steps to obtain the corresponding reward. However, I do not fully agree with this. According to equations (4) and (6), DRAIL needs to compute $\mathcal{L}\_{diff}(𝑠,𝑎,𝑐^+)$ and $\mathcal{L}\_{diff}(𝑠,𝑎,𝑐^-)$ when obtaining rewards. From equation (3), we know both terms require taking an expectation over the diffusion steps 𝑇.
>
> We would like to clarify that we use sampling to approximate the expectation in Equation 3 instead of sampling all timesteps $T$ and averaging them. In all the experiments, we sample one single denoising time step $t$ to compute the reward, and we empirically found that sampling multiple denoising time steps does not consistently improve performance.
>
> > Why does DRAIL only require two reverse steps to obtain an accurate reward? Can you provide a more detailed explanation or analysis?
>
> Our method redesigns a conditional diffusion model to learn to perform an expert/agent binary classification task. Therefore, it can naturally provide a “realness” reward according to how indistinguishable agent state-action pairs are from the expert state-action pairs. To obtain the “realness” reward in Equation 4 given an agent state-action pair, we have to compute two $\mathcal{L}_{\text{diff}}$ in Equation 3 with the positive/expert condition ($c^+$) and the negative/agent condition ($c^-$), and therefore it requires two reverse steps.
>
> >  What are the key hyperparameters for DRAIL? Can you conduct additional ablation experiments on these hyperparameters?
>
> We empirically found that our proposed method, DRAIL, is robust to hyperparameters and easy to tune, especially compared to GAIL and WAIL.
>
> Like most AIL methods, the key hyperparameters of DRAIL are the learning rates of the policy and discriminator. We additionally experimented with various values of the learning rates and reported the results in Figure R.2 in the PDF attached to the rebuttal summary. The results show that our method is robust to hyperparameter variations, including 5x, 2x, 1x, and 0.5x.
>
> We thank the reviewer for inspiring us to conduct this hyperparameter sensitivity experiment. We will revise the paper to include it.
>
> > Can you compare DAC and IQLearn in your experiments? I am very interested in seeing a comparison between the diffusion discriminative classifier and the gradient penalty.
>
> We have conducted additional experiments to address it.
>
> - **Comparison to gradient penalty**: We would like to note that one of our baselines, WAIL, has already implemented the gradient penalty and gradient clipping. As requested by the reviewer, we additionally implemented and evaluated GAIL with the gradient penalty (GAIL+GP) in CarRacing. The results are shown in Figure R.3 in the PDF attached to the rebuttal summary. GAIL with gradient penalty (GAIL+GP) initially shows a faster improvement but is unstable, and its overall performance is inferior to our method, DRAIL.
> - **Comparison to IQ-Learn**: As requested by the reviewer, we additionally implemented and evaluated IQ-Learn in CarRacing. The results in Figure R.3 show that IQ-Learn struggles at this task despite our substantial effort in experimenting with different hyperparameters, including actor learning rate ($10^{-4}$, $5 \times 10^{-5}$, $3 \times 10^{-5}$), critic learning rate ($10^{-3}$, $5 \times 10^{-4}$, $3 \times 10^{-4}$), and the entropy coefficient ($10^{-1}$, $5 \times 10^{-2}$, $2 \times 10^{-2}$, $10^{-2}$, $5 \times 10^{-3}$, $10^{-3}$), as well as trying various setups, including $\chi^2$-divergence and gradient penalty.
> - **Comparison to DAC**: The main contribution of DAC is to assign proper rewards to the “absorbing states” that an agent enters after the end of episodes. DAC requires additional annotations to determine if a state is an absorbing state or not. However, all the methods evaluated in our work do not have access to such information, making comparing them to DAC unfair. We believe the contributions of DAC and our work are orthogonal and could be combined.
>
> We will revise the paper to include the new results and the discussion.
>
> **References**
>
> [1] Wang et al. “DiffAIL: Diffusion Adversarial Imitation Learning.” In AAAI, 2024.

---

> > ### Comment · Reviewer_PZWW · 2024-08-09
> >
> > Thank you for your detailed answer to my questions. The response addressed my main concerns, therefore, I increased my score to 5.

---

> > > ### Author Response · Authors · 2024-08-10
> > > **Re: Official Comment by Reviewer PZWW**
> > >
> > > We sincerely thank the reviewer for acknowledging our rebuttal and for helping us to improve our submission.

---

### Official Review · Reviewer_3pMZ · 2024-07-11

**Soundness:** 3
**Presentation:** 3
**Contribution:** 3
**Rating:** 6
**Confidence:** 3

**Summary:**

The paper proposes a novel imitation learning framework that integrates a diffusion model into Generative Adversarial Imitation Learning (GAIL). The primary aim is to address the instability and brittleness associated with GAIL by introducing more robust and smoother reward functions for policy learning. The authors develop a diffusion discriminative classifier to enhance the discriminator's performance and generate diffusion rewards.

Extensive experiments in various domains such as navigation, manipulation, and locomotion demonstrate that DRAIL outperforms or is competitive with existing imitation learning methods. The paper highlights DRAIL’s effectiveness, generalizability, and data efficiency, offering a significant contribution to the field of imitation learning.

**Strengths:**

1. **Innovative Integration**: The integration of diffusion models into GAIL is innovative and addresses the common issue of instability in adversarial imitation learning.
2. **Comprehensive Experiments**: The authors conduct extensive experiments across diverse domains, including navigation, manipulation, and locomotion, which provide robust evidence of DRAIL's effectiveness.
3. **Generalizability and Data Efficiency**: The paper demonstrates superior generalizability to unseen states and goals, as well as high data efficiency, making the approach practical for real-world applications.
4. **Robust Reward Mechanism**: The diffusion discriminative classifier enhances the robustness of the reward functions, contributing to more stable and reliable policy learning.
5. **Visualizations**: The visualized comparisons of learned reward functions between GAIL and DRAIL effectively illustrate the advantages of the proposed approach.

###

**Weaknesses:**

1. **Complexity**: The introduction of diffusion models adds complexity to the framework, which may pose challenges in implementation and require significant computational resources.
2. **Limited Scope in Real-World Applications**: While the experiments are diverse, the applicability in highly dynamic and unpredictable real-world environments is not thoroughly explored.
3. **Comparative Analysis Depth**: Although the paper compares DRAIL with several baselines, a deeper analysis of why certain methods perform better in specific tasks could enhance understanding.
4. **Scalability**: The scalability of DRAIL in larger and more complex environments is not extensively evaluated, which could be a limitation for broader adoption.
5. **Hyperparameter Sensitivity**: The sensitivity of DRAIL to various hyperparameters is not discussed, which could impact its robustness across different settings.

###

**Questions:**

1. How does DRAIL handle environments with rapidly changing dynamics, and what are the limitations in such scenarios?
2. What are the computational overheads associated with incorporating diffusion models, and how does it compare with other state-of-the-art methods in terms of efficiency?
3. How sensitive is DRAIL to the choice of hyperparameters, and what guidelines can be provided for tuning these parameters in different environments?
4. Can the diffusion discriminative classifier be extended or modified to further improve the reward robustness and policy performance in more complex tasks?
5. What specific strategies were employed to ensure the stability of the diffusion model training, and how do these strategies affect the overall performance of DRAIL?

**Limitations:**

Shown in weakness

---

> ### Author Rebuttal · Authors · 2024-08-07
>
> We sincerely thank the reviewer for the thorough and constructive comments. Please find the response to your questions below.
>
> > Complexity: The introduction of diffusion models adds complexity to the framework, which may pose challenges in implementation and require significant computational resources. What are the computational overheads associated with incorporating diffusion models, and how does it compare with other state-of-the-art methods in terms of efficiency?
>
> We would like to clarify that our proposed method does not require significant computational resources, as it only requires two feedforward passes to obtain a reward for a state-action pair. This is because our design does not require going through the diffusion model generation process. Our method exhibits better efficiency compared to the most widely adopted method, GAIL, performing significantly better than GAIL with the same number of feedforward passes computed.
>
> > Limited Scope, Scalability, and rapidly changing dynamics
>
> This paper aims to propose a fundamental algorithm that improves the AIL framework. While previous AIL papers, such as GAIL, WAIL, and DiffAIL, have only tested their algorithms in locomotion environments, we evaluate our proposed method in various domains, including locomotion (Walker and AntReach), navigation (Maze and AntReach), robot arm manipulation (FetchPush and FetchPick), robot arm dexterity (HandRotate), and games (CarRacing). We believe this sufficiently demonstrates the broad applicability of our proposed method.
>
> >  A deeper analysis of why certain methods perform better in specific tasks could enhance understanding
>
> We thank the reviewer for the suggestion. We will provide deeper analyses of the performance of different methods on all the tasks in the revised paper.
>
> > How sensitive is DRAIL to the choice of hyperparameters, and what guidelines can be provided for tuning these parameters in different environments?
>
> We empirically found that our proposed method, DRAIL, is robust to hyperparameters and easy to tune, especially compared to GAIL and WAIL.
>
> Like most AIL methods, the key hyperparameters of DRAIL are the learning rates of the policy and discriminator. We additionally experimented with various values of the learning rates of the policy and discriminator and reported the results in Figure R.2 in the PDF attached to the rebuttal summary. The results show that our method is robust to hyperparameter variations, including 5x, 2x, 1x, and 0.5x.
>
> We thank the reviewer for inspiring us to conduct this hyperparameter sensitivity experiment. We will revise the paper to include it.
>
> > Can the diffusion discriminative classifier be extended or modified to further improve the reward robustness and policy performance in more complex tasks?
>
> We empirically observe that the rewards produced by our diffusion model classifier are robust, resulting in stable policy learning curves, as demonstrated in the experiments. It is potentially beneficial to incorporate the recent advancements in developing diffusion models into DRAIL, which is left for future work.
>
> > What specific strategies were employed to ensure the stability of the diffusion model training, and how do these strategies affect the overall performance of DRAIL?
>
> We empirically observe that the training of diffusion models is very stable with smoothly decreased losses. We believe this justifies the design of our BCE loss.

---

> ### Author Response · Authors · 2024-08-10
> **Reminder: The reviewer-author discussion period ends in three days**
>
> We would like to express our sincere gratitude to the reviewer for the thorough and constructive feedback. We are confident that our responses adequately address the concerns raised by the reviewer, including the following points.
> - A description of computational resources
> - A discussion of the broad applicability of our proposed method
> - An additional hyperparameter sensitivity experiment
> - A discussion of further improving the robustness and stability of our proposed method
>
> Please kindly let us know if the reviewer has any additional concerns or if further experimental results are required. We are fully committed to resolving any potential issues, should time permit. Again, we thank the reviewer for all the detailed review and the time the reviewer put into helping us to improve our submission.

---

> > ### Author Response · Authors · 2024-08-13
> > **Reminder: The reviewer-author discussion period ends in 20 hours**
> >
> > We would like to express our sincere gratitude to the reviewer for the thorough and constructive feedback. We are confident that our responses and additional experimental results adequately address the concerns raised by the reviewer. Please consider our rebuttal and kindly let us know if the reviewer has any additional concerns.

---

### Official Review · Reviewer_uNxe · 2024-07-22

**Soundness:** 3
**Presentation:** 3
**Contribution:** 2
**Rating:** 5
**Confidence:** 3

**Summary:**

Imitation learning (IL) is a research area that focuses on learning a policy from expert demonstrations. One of the most popular imitation learning techniques, General Adversarial Imitation Learning (GAIL), has been proposed to mitigate some of the issues that naive IL algorithms suffer from. Although GAIL has seen a lot of success, it is known to be very unstable and difficult to tune. The authors propose Diffusion Reward Adversarial Imitation Learning (DRAIL) to mitigate some of the difficulties of training GAIL. In particular, the authors use a diffusion model to train the reward function. With this change in reward function design, the authors showed in practice that policies trained with DRAIL performed better than past approaches that attempted to mitigate GAIL issues. Additionally, the authors demonstrated that they could learn a reward function with a very low number of expert trajectories compared to other approaches.

**Strengths:**

- The paper was well written, and the author's contribution was easy to follow.
- The authors performed a thorough empirical investigation to show how their proposed approach compared against baselines.
- The authors propose an algorithm that takes full advantage of recent advancements in generative modeling.

**Weaknesses:**

- The paper needs more novelty. DiffAIL proposed using diffusion as a reward, and DiffAIL also proposed using the training error as a reward signal.
- The paper needs to include comparisons to the two missing cites [1] and [2].
- The authors do not provide any insight into why the proposed reward function is better than DiffAIl, especially since DIffAIL sometimes matches or performs better than the proposed algorithms.

**Questions:**

- What denoise step values did you try? And how did they perform?
- Could you explain why the proposed reward model performs better than DiffAIL?
- If BC performs better than all the baselines on Walker, then that means the task should be easy for all the baselines. - Did you try increasing or decreasing the number of expert samples used for learning? Are you using D4RL? If so, what dataset split are you using?
- Did you try training various levels of optimal demonstration, such as walker2d-expert-v0, walker2d-medium-v0, and walker2d-random-v0, to see how well the algorithms perform when noise is introduced?
- Are you taking the argmax or sampling from the learned imitation learning policies?
- Did you try other divergences besides Jensen-Shannon?
Missing citations [1], [2],


[1] A Coupled Flow Approach to Imitation Learning by Freund 2023

[2] Diffusion Model-Augmented Behavioral Cloning by Chen et al. 2024

**Limitations:**

Yes

---

> ### Author Rebuttal · Authors · 2024-08-07
>
> We sincerely thank the reviewer for the thorough and constructive comments. Please find the response to your questions below.
>
> > The paper needs more novelty. DiffAIL proposed using diffusion as a reward, and DiffAIL also proposed using the training error as a reward signal.
>
> > ​​The authors do not provide any insight into why the proposed reward function is better than DiffAIl, especially since DIffAIL sometimes matches or performs better than the proposed algorithms.
>
> > Could you explain why the proposed reward model performs better than DiffAIL?
>
> We explicitly discuss how our proposed method differs from DiffAIL and why our proposed method produces better rewards in Section A. We will reorganize the paper to make this clear from the main paper.
>
> DiffAIL uses how well a diffusion model can denoise a state-action pair to indicate the “realness” of the pair for adversarial imitation learning. To this end, DiffAIL learns an unconditional diffusion model to denoise expert state-action pairs well, while denoise agent state-action pairs poorly. In contrast, we propose to directly train an expert/agent binary classifier by optimizing the binary cross-entropy (BCE) loss according to the denoising performance of a conditional diffusion model conditioning on expert/agent label, and our novelty lies in such a design. Our method significantly differs from DiffAIL since we formulate distinguishing expert and agent state-action pairs as a binary classification task instead of a denoising task, and this design aligns better with the GAIL formulation compared to DiffAIL. We extensively discuss this in Section A in the submission. We will revise the paper to highlight this discussion in the main paper to avoid confusion.
>
> Moreover, DiffAIL’s evaluation only considers locomotion tasks, while our work extensively compares our method in various domains, including navigation (Maze and AntReach), locomotion (Walker and AntReach), robot arm manipulation (FetchPush and FetchPick), robot arm dexterous (HandRotate), and games (CarRacing). Additionally, we present experimental results on generalization to unseen states and goals and on varying amounts of expert data.
>
> > The paper needs to include comparisons to the two missing cites [1] and [2].
>
> We thank the reviewer for providing these references. We will revise the paper to include the following discussions and cite these works.
>
> Freund et al. (2023) [1] introduce CFIL, which employs normalizing flows for state and state-action distribution modeling in imitation learning. Chen et al. (2024) [2] augment behavioral cloning using a diffusion model focusing on offline imitation learning. In contrast, AIL and our method aim to leverage online environment interactions. Also, we would like to note that Chen et al. (2024) [2] was published at ICML 2024 (July 2024), which is later than the NeurIPS 2024 submission deadline (May 2024).
>
> > What denoise step values did you try? And how did they perform?
>
> As mentioned in Section E.1, we set the total timesteps $T$ to 1000 in all experiments. During the training phase, we uniformly randomly sample the denoising steps from the range 0 to T. During the inference phase, we followed the same procedure as in the training phase, i.e., uniformly randomly sampling from [0, T].
>
> As requested by the reviewer, we additionally experimented with setting the denoising step to constant values \{250, 500, 750\}. The result is presented in Figure R.1 in the PDF attached to the overall response. The result shows that following the same denoising step sampling procedure as in the training phase, as adopted in our method, achieved the best performance. Also, consistently sampling with a large denoise step (750) hurts the policy learning performance.
>
>
> > Did you try increasing or decreasing the number of expert samples used for learning in Walker? Are you using D4RL?
>
> As mentioned in Section 5.1 (line 228), the expert Walker dataset was collected from a PPO expert policy instead of D4RL datasets. We evaluate and discuss the data efficiency, i.e., the amount of expert data required, of all the methods in Section 5.5. Figure 6 shows that our proposed method can learn with fewer demonstrations compared to the baselines in Walker and FetchPush.
>
> > Did you try training various levels of optimal demonstration, such as walker2d-expert-v0, walker2d-medium-v0, and walker2d-random-v0, to see how well the algorithms perform when noise is introduced?
>
> We didn’t try training with various levels of optimal demonstration in Walker. We evaluate the ability to generalize to varying levels of noise in Section 5.4 in FetchPush by randomizing initial states and goal locations. The results in Figure 5 show that our proposed method demonstrates the highest robustness towards noisy environments.
>
> > Are you taking the argmax or sampling from the learned imitation learning policies?
>
> We use PPO as our RL algorithm to learn the agent policy. Hence, the policy is stochastic during training and deterministic during inference.
> > Did you try other divergences besides Jensen-Shannon?
>
> We use JS divergence following the GAIL’s original setting. Exploring other divergences and distance metrics, such as the Wasserstein distance or f-divergences, is a promising future direction. We thank the reviewer for the suggestion and will revise the paper to include this discussion.
>
> **References**
>
> [1] Freund et al. “A Coupled Flow Approach to Imitation Learning.” In PMLR, 2023.
>
> [2] Chen et al. “Diffusion Model-Augmented Behavioral Cloning.” In ICML, 2024.

---

> ### Author Response · Authors · 2024-08-10
> **Reminder: The reviewer-author discussion period ends in three days**
>
> We would like to express our sincere gratitude to the reviewer for the thorough and constructive feedback. We are confident that our responses adequately address the concerns raised by the reviewer, including the following points.
> - A detailed description of our contributions compared to DiffAIL
> - An intuitive explanation of why our method works
> - A discussion of Freund et al., 2023 (CFIL) and Chen et al., 2024 (DBC)
> - A description of data efficiency experiments, i.e., learning from different amounts of expert data, provided in the main paper
> - An explanation of training and sampling from learned policies using PPO
> - A discussion of using different divergence measures
>
> Please kindly let us know if the reviewer has any additional concerns or if further experimental results are required. We are fully committed to resolving any potential issues, should time permit. Again, we thank the reviewer for all the detailed review and the time the reviewer put into helping us to improve our submission.

---

> > ### Author Response · Authors · 2024-08-13
> > **Reminder: The reviewer-author discussion period ends in 20 hours**
> >
> > We would like to express our sincere gratitude to the reviewer for the thorough and constructive feedback. We are confident that our responses and additional experimental results adequately address the concerns raised by the reviewer. Please consider our rebuttal and kindly let us know if the reviewer has any additional concerns.

---

### Author Rebuttal · Authors · 2024-08-07

The attached PDF file contains the following content:
- **[Reviewer uNxe] The Effect of Denoising Time Step**: We experimented with computing rewards using different constant denoising time steps and reported the result in Figure R.1. The result shows that following the same denoising step sampling procedure as in the training phase, i.e., uniformly randomly sample from [0, T], as adopted in our method, achieves the best performance, justifying our design choice.
- **[Reviewer 3pMZ, Reviewer PZWW] Hyperparameter Ablation Study**:  We experimented with varying hyperparameters, including the discriminator learning rate ($\eta_\phi$) and policy learning rates ($\eta_\pi$). The results are shown in Figure R.2, demonstrating that DRAIL maintains robust performance with varying hyperparameters.
- **[Reviewer PZWW] Comparison to Gradient Penalty and IQ-Learn**: We implemented and evaluated two additional baselines, GAIL with Gradient Penalty and IQ-Learn, as suggested by the reviewer. The results shown in Figure R.3 highlight the superior performance of our proposed method.

---

### Decision · Program_Chairs · 2024-09-25

**Decision:**

Accept (poster)

**Comment:**

This paper proposes a diffusion-based adversarial imitation learning method to mitigate the instability of the prior work, generative adversarial imitation learning (GAIL). All of the reviewers found that the empirical result across a variety of benchmarks is strong enough to show that the proposed method is robust and sample-efficient. Although there were questions and concerns regarding the position of the work relative to the previous work and the lack of certain baselines, the authors clarified them and added additional baselines during the rebuttal period. As a result, all of the reviewers acknowledged that their major concerns were addressed. Assuming that the authors will make the contribution clearer in the camera-ready version, I recommend to accept this paper.